# PHASE-AWARE TRAINING SCHEDULE SIMPLIFIES LEARNING IN FLOW-BASED GENERATIVE MODELS

## ABSTRACT

We analyze the training of a two-layer autoencoder used to parameterize a flow-based generative model for sampling from a high-dimensional Gaussian mixture. Previous work shows that the phase where the relative probability between the modes is learned disappears as the dimension goes to infinity without an appropriate time schedule. We introduce a time dilation that solves this problem. This enables us to characterize the learned velocity field, finding a first phase where the probability of each mode is learned and a second phase where the variance of each mode is learned. We find that the autoencoder representing the velocity field learns to simplify by estimating only the parameters relevant to each phase. Turning to real data, we propose a method that, for a given feature, finds intervals of time where training improves accuracy the most on that feature. Since practitioners take a uniform distribution over training times, our method enables more efficient training. We provide preliminary experiments validating this approach.

## 1 INTRODUCTION

In recent years, diffusion models have emerged as a powerful technique for learning to sample from high-dimensional distributions Sohl-Dickstein et al. (2015); Song et al. (2021); Song & Ermon (2020); Ho et al. (2020), especially in the context of generating images and recently also for text Lou et al. (2024). The idea lies in learning, from data samples, a velocity field that pushes noisy datapoints to clean datapoints. Despite the remarkable performance of these models, there remain several open questions, including understanding what makes a good noise schedule, which is the focus of this paper.

We consider the problem of training a neural network to learn the velocity field to generate samples from a two-mode Gaussian mixture (GM). This serves as a prototypical example to understand how diffusion models handle learning features at different scales, since the two-mode GM has two scales: the macroscopic scale of the probability of each mode, and the microscopic scale of the variance of each mode.

This problem was previously considered by Cui et al. (2024), but their analysis only handles the *balanced* two-mode GM (i.e. the probability of each mode is exactly $1/2$.) On the other hand, Biroli et al. (2024) assume access to the exact velocity field and find that the phase where the probability of each mode is learned disappears as the dimension of the problem grows.

In this work, we first introduce a noise schedule that makes the phase where this probability is learned not disappear as the dimension goes to infinity. This enables us to extend the analysis of Cui et al. (2024) to the two-mode GM without the balanced assumption. More precisely, our contributions are as follows.

- We give an asymptotic characterization of the learned velocity field for learning to generate the two-mode GM, finding a separation into two phases. We further show that $\Theta_d(1)$ samples are sufficient to learn the velocity field.

- We show that the neural network representing the velocity field learns to simplify for each phase. In the first phase, it only concerns estimation of the probability of each mode, whereas in the second phase, it concerns estimation of the variance of each mode. This sheds light on the advantage of diffusion models over denoising autoencoders, since the

sequential nature of diffusion models shown here allows them to decompose the complexity of the problem.

- We show that the phase transition separating the two phases can be detected from a discontinuity in the Mean Squared Error associated to the learning problem, which suggests a way to find these transitions for general data distributions.

- For real data, this analysis suggests that training more at the times associated with a feature improves accuracy on that feature. In fact, we propose a method that, given a feature, finds an interval of time where more training improves accuracy on that feature the most. We further validate this on the MNIST dataset. We provide the code for the experiments here.

## 2 RELATED WORKS

**Phase transitions of generative models in high dimensions.** Several works analyze phase transitions in the dynamics of generative models. Raya & Ambrogioni (2023) find that diffusion models can exhibit symmetry breaking, where two phases are separated by a time where the potential governing the dynamics has an unstable fixed point. They give a full theoretical analysis for the data being two equiprobable point masses in $\mathbb{R}$, and also give a bound for the symmetry breaking time for the case where the data is a sum of finitely many point masses. Our setting generalizes the case of two equiprobable point masses in $\mathbb{R}$ to two Gaussians in $\mathbb{R}^d$ that are not necessarily equiprobable. Ambrogioni (2023) builds on Raya & Ambrogioni (2023) and shows several connections between equilibrium statistical mechanics and the phase transitions of diffusion models. Ambrogioni (2023) further conjectures that accurately sampling near times of "critical generative instability" affects the sample diversity. We give an explicit description of these critical times and verify this conjecture theoretically for sampling (see Proposition 1) and for learning (see Corollary 5) and empirically for learning (see Section 6). Li & Chen (2024) also formalize the study of critical windows taking the data to be a mixture of strongly log-concave densities. They give non-asymptotic bounds for the start and end times of these critical windows, which have a closed form expression for the mixtures of isotropic Gaussians case. In contrast, we provide sharp asymptotic characterizations for the phase transition times. Biroli & Mézard (2023) analyze the Curie-Weiss model and analytically characterize the speciation time, defined as the time after which the mode that the sample will belong to is determined. Biroli et al. (2024) generalize the result and find an speciation time $t_s \sim \frac{1}{2} \log(\lambda)$ for an Ornstein-Uhlenbeck process where $\lambda$ is the largest eigenvalue of the covariance of the data, usually proportional to $d$. Montanari (2023) points out a similar phase transition when *learning* the velocity field to generate from a two-mode unbalanced Gaussian mixture, leading to problems for accurate estimation of the data. Montanari (2023) addresses this by using a different neural network to learn each mode. In the current work, we show that it is not necessary to tailor the network for each mode if the right time schedule is used. It is worth noting that all these works are about sampling. We provide a result for sampling in Proposition 1. Building on this, we give results for learning (i.e. estimating the velocity field through a neural network) which is the main contribution of our paper.

**Time-step complexity.** Several results give convergence bounds detailing the required time-steps, score accuracy, and/or data distribution regularity to sample accurately. Benton et al. (2024) show that at most $O(d \log^2(1/\delta)/\epsilon^2)$ time steps are required to approximate a distribution corrupted with Gaussian noise of variance $\delta$ to within $\epsilon^2$ KL divergence. Chen et al. (2023) study probability flow ODE and obtain $O(\sqrt{d})$ convergence guarantees with a smoothness assumption. An underlying assumption in all these works is that the score or velocity field is learned to a certain accuracy. In the present work, we address this problem in the special case of a Gaussian mixture.

**Sample complexity for Gaussian mixtures.** Cui et al. (2024) study the learning problem for the Gaussian mixture in high dimensions and demonstrate that $n = \Theta_d(1)$ samples are sufficient in the balanced case where the two modes have the same probability. This is done through statistical physics techniques of computing the partition function and using a sample symmetric ansatz. As we show, due to the speciation time at $d^{-1/2}$ which tends to zero as the dimension $d$ grows, this analysis misses one phase of learning. Gatmiry et al. (2024) show that quasi-polynomial $(O(d^{\text{poly}(\log(\frac{d+k}{\epsilon}))}))$ sample and time complexity is enough for learning $k$-gaussian mixtures. The data distribution is more general than the one we consider, but on the other hand we give a $\Theta_d(1)$ sample and time complexity.

## 3 BACKGROUND

**Data and flow-based generative model.** Consider the two-mode Gaussian mixture (GM)

$$\rho = p\mathcal{N}(\mu, \sigma^2 \text{Id}_d) + (1-p)\mathcal{N}(-\mu, \sigma^2 \text{Id}_d) \tag{1}$$

where $p \in (0,1)$ and $\mu \in \mathbb{R}^d$ such that $\|\mu\|^2 = d$ and $\sigma = \Theta_d(1)$. A diffusion model for $\rho$ starts with samples from a simple distribution (say a Gaussian) and sequentially denoises them to get samples from the data. More precisely, consider the stochastic interpolant

$$x_t = \alpha_t x_0 + \beta_t x_1 \tag{2}$$

where $x_0 \sim \mathcal{N}(0, \text{Id}_d)$, $x_1 \sim \rho$, and $\alpha_t, \beta_t : [0,1] \to \mathbb{R}$, $\alpha_0 = 1 = \beta_1$, $\alpha_1 = 0 = \beta_0$. Stochastic interpolants are introduced in Albergo et al. (2023), and they prove that if $X_t$ solves the probability flow ODE

$$\dot{X}_t = b_t(X_t) \qquad \text{with} \qquad b_t(x) = \mathbb{E}[\dot{x}_t | x_t = x] \tag{3}$$

with $X_0 \sim \mathcal{N}(0, \text{Id}_d)$, we then have $X_t \overset{d}{=} x_t$ for $t \in [0,1]$ and hence $X_{t=1} \sim \rho$. We call $X_t$ the flow-based generative model associated to the interpolant $I_t$.

Since $\rho$ is a Gaussian mixture, the expression for the exact velocity field $b_t(x)$ from equation 3 can be computed exactly. Our goal is to understand how well a neural network can estimate this velocity field through samples, in the large dimension $d \to \infty$ limit assuming low sample complexity for the data $n = \Theta_d(1)$.

**Loss function.** To fulfill our goal, we rewrite the velocity field as

$$b_t(x) = \left( \dot{\beta}_t - \frac{\dot{\alpha}_t}{\alpha_t}\beta_t \right) f(x,t) + \frac{\dot{\alpha}_t}{\alpha_t}x, \tag{4}$$

where $f(x,t) = \mathbb{E}[x_1 | x_t = x]$ is called the denoiser since it recovers the datapoint $x_1$ from a noisy version $x_t$. The denoiser is characterized as the minimizer of the loss (see Albergo et al. (2023))

$$\mathcal{R}[f] = \int_0^1 \mathbb{E}\|f(x_t, t) - x_1\|^2 dt. \tag{5}$$

In practice, however, we usually do not have access to the exact data distribution. So we assume we have a dataset $\mathcal{D} = \{x_1^\mu\}_{\mu=1}^n$ where $x_1^\mu \sim_{\text{iid}} \rho$. On the other hand, we have unlimited samples from $x_0 \sim \mathcal{N}(0, \text{Id}_d)$. Hence, to each data sample $x_1^\mu$ we can associate several noise samples $x_0^{\mu,\nu}$ with $\nu = 1, \cdots, k$. We then denote $x_t^{\mu,\nu} = \alpha_t x_0^{\mu,\nu} + \beta_t x_1^\mu$. Later in our analysis, we will assume infinitely many noise samples associated to each data sample, so that we can take expectation with respect to the noise distribution.

We parameterize the denoiser with a single neural network for each $t$, which we denote as $f_{\theta_t}(x)$. We get then an empirical version of the loss in equation 5 $\hat{\mathcal{R}}(\{\theta_t\}_{t \in [0,1]}) = \int_0^1 \hat{\mathcal{R}}_t(\theta_t)dt$ where

$$\hat{\mathcal{R}}_t(\theta_t) = \sum_{\mu=1}^n \sum_{\nu=1}^k \|f_{\theta_t}(x_t^{\mu,\nu}) - x_1^\mu\|^2 \tag{6}$$

**Network architecture.** We focus on the case where the neural network parameterizing the denoiser function $f(x,t)$ is a two-layer denoising autoencoder with a trainable skip connection as follows

$$f_{\theta_t}(x) = c_t x + u_t \tanh\left( \frac{w_t \cdot x}{\sqrt{d}} + b_t \right) \tag{7}$$

where $\theta_t = \{c_t, u_t, w_t, b_t\}$; $c_t, b_t \in \mathbb{R}$; and $u_t, w_t \in \mathbb{R}^d$. The structure of this denoising autoencoder is a particular case of the U-Net from Ronneberger et al. (2015) and is motivated by the exact denoiser which can be computed exactly since the data distribution is a Gaussian mixture

$$\mathbb{E}[x_1 | x_t = x] = \frac{\beta_t \sigma^2}{\alpha_t^2 + \sigma^2 \beta_t^2}x + \frac{\alpha_t^2}{\alpha_t^2 + \sigma^2 \beta_t^2}\mu \tanh\left( \frac{\beta_t}{\alpha_t^2 + \sigma^2 \beta_t^2}\mu \cdot x + h \right) \tag{8}$$

where $h$ is such that $e^h/(e^h + e^{-h}) = p$. (See Albergo et al. (2023), Appendix A for the proof.)

We add to the loss regularization terms for $w_t$ and $u_t$, giving

$$\hat{\mathcal{R}}_t(\theta_t) = \sum_{\mu=1}^{n} \sum_{\nu=1}^{k} ||f_{\theta_t}(x_t^{\mu,\nu}) - x_1^{\mu}||^2 + \frac{\lambda}{2}||u_t||^2 + \frac{\ell}{2}||w_t||^2 \tag{9}$$

Denoting $\hat{\theta}_t$ the minimizer of this loss, we define

$$\hat{b}_t(x) = \left( \dot{\beta}_t - \frac{\dot{\alpha}_t}{\alpha_t}\beta_t \right) f_{\hat{\theta}_t}(x) + \frac{\dot{\alpha}_t}{\alpha_t}x. \tag{10}$$

Using this velocity field, we then run the probability flow ODE

$$\dot{\hat{X}}_t = \hat{b}_t(\hat{X}_t); \quad \hat{X}_0 \sim \mathcal{N}(0, \mathrm{Id}_d). \tag{11}$$

Our goal is to understand how close $\hat{X}_2$ is to a sample from the Gaussian mixture $\rho$.

Cui et al. (2024) consider the special case of tied weights $u_t = w_t$ and $b_t = 0$. This is enough to learn to sample from the balanced two-mode GM (i.e. $p = 1/2$) but fails at the two-mode GM for $p \neq 1/2$. This follows because $x_0$ has an even distribution and their choice of tied weights and no bias yields an odd velocity field which results in an even distribution for $x_t$. If the weights are untied and the bias is added, the analysis of Cui et al. (2024) still does not work to show that $\hat{X}_1$ has the correct $p$ for $p \neq 1/2$. This is because the gradients for $w_t$ and $b_t$ vanish as $d \to \infty$ unless special care is given to the small times where a phase transition related to learning the probability between the modes occurs, as will be explained next.

**Separation into phases.** Biroli et al. (2024) show that the generative model with the exact velocity field from equation 3 with $\alpha_t = \sqrt{1 - t^2}$ and $\beta_t = t$ undergoes a phase transition at the speciation time $t_s = 1/\sqrt{d}$. The speciation time is defined as the time in the generation process after which the mode that the sample will belong to at the end of the process is determined. Their analysis can be extended to show that the speciation time is still $t_s = 1/\sqrt{d}$ if we instead have $\alpha_t = 1 - t$ and $\beta_t = t$ which are the choices in our paper. Since this result is only mentioned as motivation, we will not prove it.

The analysis of Cui et al. (2024) relies on taking the $d \to \infty$ limit and obtaining a limiting ODE. Since $t_s = 1/\sqrt{d}$ goes to zero as $d \to \infty$, their limiting ODE has a singularity at $t = 0$ and the possibility of learning the probability of each mode is lost. This is in essence why the analysis of Cui et al. (2024) can not capture the learning of $p$ for $p \neq 1/2$.

We will dilate time so as to make the speciation time $t_s$ not disappear as $d \to \infty$. More precisely, we define

$$\tau(t) = \begin{cases} \frac{\kappa t}{\sqrt{d}} & \text{if } t \in [0, 1] \\ \frac{\kappa}{\sqrt{d}} + \left(1 - \frac{\kappa}{\sqrt{d}}\right)(t-1) & \text{if } t \in [1, 2]. \end{cases} \tag{12}$$

This fulfills $\tau(0) = 0, \tau(1) = \kappa/\sqrt{d}$, and $\tau(2) = 1$. We prove next that the generative model from equation 3 with $\alpha_t = 1 - \tau_t$ and $\beta_t = \tau_t$ has two phases: for $t \in [0, 1]$ the probability of each mode is estimated, and for $t \in [1, 2]$ the variance of each mode is estimated.

**Proposition 1.** *Let $X_t$ be the solution to the probability flow ODE from equation 3 with $\alpha_t = 1 - \tau_t$ and $\beta_t = \tau_t$ where $\tau_t$ is defined in equation 12. Then for $t \in [0, 2]$ we have*

$$X_t - \frac{\mu \cdot X_t}{d}\mu \sim \mathcal{N}\left(0, \sigma_t^2 Id_{d-1}\right).$$

*where $\sigma_t$ is characterized below. We further have the following phases*

- ***First phase:** For $t \in [0, 1]$, we have $\lim_{d \to \infty} \sigma_t = 1$.*

  *In addition, $\nu_t = \lim_{d \to \infty} \frac{\mu \cdot X_t}{\sqrt{d}}$ fulfills*

  $$\nu_1 \sim p\mathcal{N}(\kappa, 1) + (1-p)\mathcal{N}(-\kappa, 1).$$

- **Second phase**: *We have* $\lim_{d\to\infty} \sigma_2 = \sigma$.

  *In addition,* $M_t = \lim_{d\to\infty} \frac{\mu \cdot X_t}{d}$ *fulfills*

  $$M_2 \sim p_\kappa \delta_1 + (1 - p_\kappa)\delta_{-1}$$

  *where* $p^\kappa$ *is such that* $\lim_{\kappa\to\infty} p_\kappa = p$

See Appendix A for the proof of this Proposition. In Appendix E, we give a generalization of the time dilation formula in equation 12 for a Gaussian mixture with more than two modes.

Without the time dilation, we can not capture the learning of $p$ for $p \neq 1/2$ because the first phase (where this parameter is learned) disappears as $d \to \infty$. The time dilation will allow us to analyze the phase where $p$ is learned in the $d \to \infty$ limit and hence show that $\hat{X}_2$ recovers $p$.

We show this in two steps. In Section 4, we characterize the learned parameters of the velocity field in terms of a few projections, called the overlaps. Then, in Section 5, we combine these characterizations with Proposition 1 to show that $\hat{X}_2$ recovers the parameters $p$ and $\sigma^2$ of the two-mode Gaussian mixture $\rho$ under appropriate limits.

## 4 LEARNING

In this section, we will characterize $\hat{\theta}_t$, the minimizer of the loss from equation 9, which is used to parameterize the velocity field that yields $\hat{X}_t$ (see equation 11.) We take $\alpha_t = 1 - \tau_t$ and $\beta_t = \tau_t$ and analyze $\hat{\theta}_t$ in the $d \to \infty$ limit. We first analyze the times $t \in [0, 1]$ and then $t \in [1, 2]$.

### 4.1 FIRST PHASE

The interpolant from equation 2 in the first phase reads

$$x_t^\mu = \left(1 - \frac{\kappa t}{\sqrt{d}}\right) x_0^\mu + \frac{\kappa t}{\sqrt{d}} x_1^\mu$$

where $t \in [0, 1]$. To characterize $\hat{\theta}_t = \{c_t, u_t, w_t, b_t\}$, we introduce the following overlaps (dropping the dependence on $t$ for notational simplicity.)

$$p_\eta^\mu = \frac{z^\mu \cdot w}{d} \quad \omega = \frac{\mu \cdot w}{d} \quad r = \frac{\|w\|^2}{d} \quad q_\xi^\mu = \frac{x_0^\mu \cdot u}{d} \quad q_\eta^\mu = \frac{z^\mu \cdot u}{d} \quad m = \frac{\mu \cdot u}{d} \quad q = \frac{\|u\|^2}{d}. \quad (13)$$

We now give equations for the overlaps in the asymptotic $d \to \infty$ limit.

**Result 1** (Sharp Characterization of Parameters in First Phase). *For any $t \in [0, 1]$, the overlaps associated to $\hat{\theta}_t$, the minimizer of the loss from equation 9, satisfy the following in the $d \to \infty$ limit*

$$q_\eta = \frac{\sigma\overline{\phi}}{\lambda + n\overline{\phi^2}} \qquad\qquad c = q_\xi = p_\eta = 0$$

$$\qquad\qquad\qquad\qquad q = m^2 + nq_\eta^2$$

$$m = \frac{n\overline{\phi s}}{\lambda + n\overline{\phi^2}} \qquad\qquad r = \omega^2$$

$$(\lambda + n\overline{\phi^2})(\sigma(\overline{\phi'})(\overline{\phi}) + n(\overline{\phi's})(\overline{\phi s})) = (n^2\overline{\phi s}^2 + n\sigma^2\overline{\phi}^2)(\overline{\phi'\phi})$$

$$\hat{r}(\lambda + n\overline{\phi^2})^2 = -n((\lambda + n\overline{\phi^2})(\sigma(\overline{\phi''})(\overline{\phi}) + n(\overline{\phi''s})(\overline{\phi s})) - (n^2\overline{\phi s}^2 + n\sigma^2\overline{\phi}^2)(\overline{\phi\phi')'})$$

$$\omega(\ell + \hat{r})(\lambda + n\overline{\phi^2})^2 = (n\kappa t)((\lambda + n\overline{\phi^2})(\sigma(\overline{\phi's})(\overline{\phi}) + n(\overline{\phi'})(\overline{\phi s})) - (n^2\overline{\phi s}^2 + n\sigma^2\overline{\phi}^2)(\overline{\phi'\phi s}))$$

*Here and in what follows, we denote*

$$\overline{y} = \frac{1}{nk}\sum_{\mu=1}^{n}\sum_{\nu=1}^{k}\mathbb{E}_{z^{\mu,\nu}}[y^{\mu,\nu}] = \overline{p}\mathbb{E}_{z^{\mu,\nu}}[y^{\mu,\nu}|s^\mu = 1] + (1 - \overline{p})\mathbb{E}_{z^{\mu,\nu}}[y^{\mu,\nu}|s^\mu = -1].$$

See Appendix B.1 for a heuristic derivation of this result, at the level of rigor of theoretical physics. We next show that the equations for the overlaps simplify in the $n \to \infty$ limit.

**Corollary 1** (Parameters given infinite samples). *For any $t \in [0, 1]$, taking $d \to \infty$ and then $n \to \infty$ gives the following overlaps*

$$\tanh(b) = 2\left(p - \tfrac{1}{2}\right), \qquad\qquad m = 1,$$
$$c = q_\xi = q_\eta = p_\eta = 0, \qquad\qquad \omega = \kappa t.$$

See Appendix B.1.1 for the derivation. Note that the overlaps in the $n \to \infty$ limit do not contain any information about $\sigma^2$, showing that the estimation of $\sigma^2$ happens completely in the second phase.

We now turn to the Mean Squared Error. Define the scaled train and test MSE of the denoiser as

$$\text{mse}_{\text{train}} = \frac{1}{dnk} \sum_{\mu=1}^{n} \sum_{\nu=1}^{k} ||f_{\theta_t}(x_t^{\mu,\nu}) - x_1^\mu||^2 \qquad \text{mse}_{\text{test}} = \frac{1}{d}\mathbb{E}\left[||f_{\hat{\theta}_t}(x_t) - x_1||^2\right].$$

Using the above results we characterize the MSE

**Corollary 2.** *In the limit of $d \to \infty$,*

$$mse_{train} = 1 + \sigma^2 + c^2 + q\overline{\phi^2} - 2\overline{s\phi}(m + \sigma q_\eta - cq_\xi)$$
$$mse_{test} = 1 + \sigma^2 + c^2 + q\overline{\phi^2} - 2\overline{s\phi}m$$

*For $n \to \infty$, we get*

$$mse_{train} = mse_{test} = \sigma^2 + (1 - \overline{\phi s}).$$

## 4.2 SECOND PHASE

We now consider times $t \in [1, 2]$ which means we have

$$x_t^\mu = (2 - t)\left(1 - \frac{\kappa}{\sqrt{d}}\right)x_0^\mu + \left(\frac{\kappa}{\sqrt{d}} + \left(1 - \frac{\kappa}{\sqrt{d}}\right)(t-1)\right)x_1^\mu.$$

Using the same definitions of overlaps as for the first phase, we find closed-form equations for the overlaps in the asymptotic $d \to \infty$ limit, and again find the limit as $n \to \infty$ for the overlaps. See Appendix B.2 for a heuristic derivation of this result

**Result 2** (Sharp Characterization of Parameters in Second Phase). *For any $t \in [1, 2]$, in the $d \to \infty$ limit, the parameters minimizing the loss from equation 9 satisfy the following equations*

$$q_\xi = \frac{c(1 - \tau)}{\lambda + n} \qquad\qquad m = \frac{n(1 - c\tau)}{\lambda + n}$$
$$q_\eta = \frac{\sigma(1 - c\tau)}{\lambda + n} \qquad\qquad q = m^2 + nq_\xi^2 + n\sigma^2 q_\eta^2$$

$$c = \frac{\tau\left((1 + \sigma^2)(\lambda + n) - (\sigma + n)\right)}{(\lambda + n)((1 - \tau^2) + (1 + \sigma^2)\tau^2) + ((1 - \tau)^2 - \tau^2(\sigma + n))}$$

*where $\tau = t - 1$.*

**Corollary 3** (Parameters given inifite samples). *For any $t \in [1, 2]$, taking $d \to \infty$ and then $n \to \infty$ gives the following overlaps*

$$c = \frac{\tau\sigma^2}{1 + (\sigma^2 - 1)\tau^2} \qquad q_\xi = q_\eta = 0 \qquad m = 1 - c\tau$$

*where $\tau = t - 1$.*

In contrast to the first phase, the parameter $p$ does not appear in the overlaps whereas now $\sigma^2$ does. Hence, combining Corollaries 1 and 3 shows that the separation into phases can be learned by the generative model.

We also obtain the MSE for the second phase

**Corollary 4.** *In the limit of $d \to \infty$, we have*

$$mse_{train} = (1 + \sigma^2)(1 - c\tau)^2 + c^2(1 - \tau)^2 + q - 2(1 - c\tau)(\sigma q_\eta + m) + 2c(1 - \tau)q_\xi$$
$$mse_{test} = (1 + \sigma^2)(1 - c\tau)^2 + c^2(1 - \tau)^2 + q - 2(1 - c\tau)m$$

*For $n \to \infty$, we get*

$$mse_{train} = mse_{test} = \sigma^2(1 - c\tau)^2 + c^2(1 - \tau)^2.$$

*where $\tau = t - 1$.*

In Appendix B.2 we show that combining Corollaries 2 and 4 gives

**Corollary 5.** *Taking $d \to \infty$ then $n \to \infty$*

$$mse_{test} = \begin{cases} \sigma^2 + 4p(1 - p) & \text{if } t = 0 \\ \sigma^2 + (1 - \overline{\phi^2}) & \text{if } t \in (0, 1) \\ \sigma^2 & \text{if } t = 1^+ \\ 0 & \text{if } t = 2 \end{cases}$$

If we had not dilated time, in the limit of $d \to \infty$ and $\kappa \to \infty$ the $mse_{test}$ would have a jump from $\sigma^2 + 4p(1 - p)$ at $t = 0$ to $\sigma^2$ at $t = 0^+$. By dilating time, we make a transition between these two values with $t \in [0, 1]$ $mse_{test} = \sigma^2 + (1 - \overline{\phi^2})$ where $\overline{\phi^2}$ depends on time.

Remarkably, this result suggests a way to detect phase transitions for a general data distribution. Indeed, to detect the phase transition we could have as well ensure that the mse was continuous in the $d \to \infty$ limit. More generally, this suggests that having an mse that decreases smoothly as time grows would resolve the phase transitions present in the data. We leave the study of this conjecture to future work.

## 5 GENERATION

Having characterized the parameters $\hat{\theta}_t$, we now show that $\hat{X}_2$ has the right parameters $p$ and $\sigma^2$ from the data distribution $\rho$. Let $X_t$ be the solution to the ODE from equation 3 using the exact denoiser from equation 8. Assume $X_t$ and $\hat{X}_t$ have a shared initial condition $X_{t=0} = \hat{X}_{t=0} \sim \mathcal{N}(0, \text{Id}_d)$. Then $X_t - \hat{X}_t$ fulfills an ODE with initial condition 0 whose velocity field is in the span of $u_t$ and $\mu$.

Result 1 gives that in the first phase $q = m^2 + nq_\eta^2$. This can be explicitly stated as

$$\lim_{d \to \infty} \frac{\|u\|^2}{d} = \lim_{d \to \infty} \left( \frac{\mu \cdot u}{d} \right)^2 + \left( \frac{\eta \cdot u}{d} \right)^2$$

where $\eta = \sigma \sum_{\mu=1}^n z^\mu$. This means that $u_t$ is asymptotically contained in $\text{span}(\mu, \eta)$, in the sense that the projection to the complement of $\text{span}(\mu, \eta)$ has asymptotically vanishing norm, for $t \in [0, 1]$. Similarly, from Result 2, we get $q = m^2 + nq_\xi^2 + nq_\eta^2$, which means that $u_t$ is asymptotically contained in $\text{span}(\mu, \eta, \xi)$ for $t \in [1, 2]$ where $\xi = \sum_\mu s^\mu x_0^\mu$. This means that to show that $X_t$ is close to $\hat{X}_t$, it suffices to bound the projections of $X_t - \hat{X}_t$ onto $\mu$, $\eta$, and $\xi$. In fact, we have the following result (see Appendix C)

**Result 3.** *Let $X_t$ be the solution of the probability flow ODE from equation 3 using the exact denoiser from equation 8. Let $\hat{X}_t$ be the solution using the learned denoiser. Assume $X_{t=0} = \hat{X}_{t=0} \sim \mathcal{N}(0, Id_d)$. Then for $w \in span(\mu, \eta, \xi)$, with $\|w\|_2 = 1$, we have*

$$\lim_{d \to \infty} \frac{w \cdot (X_2 - \hat{X}_2)}{\sqrt{d}} = O\left( \frac{1}{n} \right).$$

*For $w \in span(\mu, \eta, \xi)^\perp$, with $\|w\|_2 = 1$, we have*

$$\lim_{d \to \infty} \frac{w \cdot (X_2 - \hat{X}_2)}{\sqrt{d}} = 0.$$

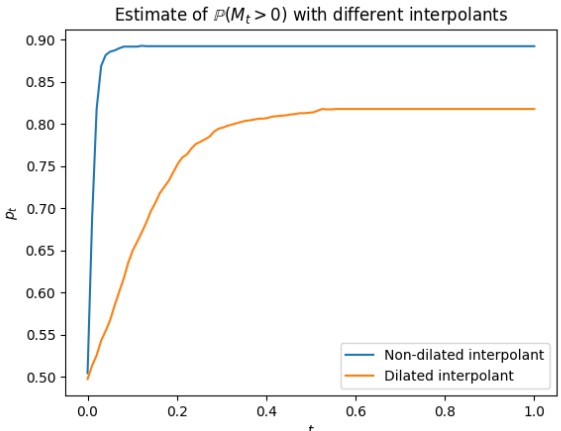

Figure 1: We learn the parameters from equation 7 for different choices of interpolant. In all experiments, we take 100 discretization points, train for 5000 epochs, with $n = 128$, $d = 5000$, and $p = .8$. We then run the probability flow ODE with the learned parameters for $K = 2000$ realizations and estimate $\mathbb{P}(M_t > 0) = p$ with $M_t = \mu \cdot X_t / d$. For the non-dilated interpolant in blue, we use $\alpha_t = 1 - t$, $\beta_t = t$. We predict the speciation to happen near $t = 1/\sqrt{5000} \approx .014$ as confirmed by the experiment since most of the speciation occurs at the first two ODE steps. For the dilated interpolant in orange, we use $\alpha_t = 1 - \tau_t$, $\beta_t = \tau_t$, $\kappa = 4$. We see the dilated interpolant estimates $p = .8$ much better than the non-dilated one.

**Corollary 6** (Parameters $p$ and $\sigma^2$ are estimated correctly). *Let $\hat{X}_t$ be the solution of the probability flow ODE from equation 3 using the learned denoiser, starting from $\hat{X}_0 \sim \mathcal{N}(0, Id_d)$. We have*

$$\lim_{\kappa \to \infty} \lim_{n \to \infty} \lim_{d \to \infty} \frac{\mu \cdot \hat{X}_2}{d} \sim p\delta_1 + (1 - p)\delta_{-1}.$$

*For $w \perp \mu$, with $\|w\|_2 = 1$, we have*

$$\lim_{n \to \infty} \lim_{d \to \infty} \frac{w \cdot \hat{X}_2}{\sqrt{d}} \sim \mathcal{N}(0, \sigma^2).$$

We conclude that the distribution generated using the learned denoiser captures both $p$ and $\sigma^2$.

## 6 EXPERIMENTS

### 6.1 VERIFICATION THAT PARAMETER $p$ IS CAPTURED

To demonstrate the difference between the time dilated and non-dilated interpolants in practice we construct the following simple experiment. We run Gradient Descent with the Adam optimizer Diederik (2014) to learn the parameters $w_t, c_t, u_t, b_t$ in equation 7 both for $\alpha_t = 1 - t$, $\beta_t = t$ and the dilated version $\alpha_t = 1 - \tau_t$, $\beta_t = \tau_t$. The results are shown in Figure 1 and suggest time-dilation is required to estimate the probability of each mode.

The code for this experiment is available here.

### 6.2 TRAINING A GIVEN FEATURE ON REAL DATA: MNIST

Recall that in the background we mentioned that the analysis of Biroli et al. (2024) shows that taking $\alpha_t = 1 - t$ and $\beta_t = t$ without any time-dilation gives an speciation time $t_s = 1/\sqrt{d}$. This then means that probability of each mode (given by $p$) can not be captured as $d \to \infty$. Our analysis then shows that if we dilate time by stretching the interval $[0, \kappa/\sqrt{d}]$ to $[0, 1]$ and the interval $[\kappa/\sqrt{d}, 1]$ to $[1, 2]$, then we get accurate estimation of $p$.

When training diffusion models in practice, we first sample a batch of times $t_1, \cdots, t_k$ uniformly. We then draw $x_0^\mu \sim \mathcal{N}(0, \mathrm{Id}_d)$, $x_1^\mu$ from our data distribution, and form a noisy sample $x_{t^\mu}^\mu = (1 - t^\mu)x_0^\mu + t^\mu x_1^\mu$ for $\mu = 1, \cdots, k$. We finally train on the loss

$$\hat{\mathcal{R}}(\theta) = \sum_{\mu=1}^{k} ||f_\theta(x_{t^\mu}^\mu, t^\mu) - x_1^\mu||^2. \tag{14}$$

where we took time as a parameter of the network as it is usually done in practice, as opposed to having a separate network for each time $t$.

*The insight of our analysis is that instead of taking the batch of times uniformly, we can sample more times near the phase transition associated to a given feature, and in this way improve accuracy on that feature.*

For a given feature, we can find the times where that feature is learned using the U-Turn method (Sclocchi et al. (2024), Biroli et al. (2024)). Consider a dataset where each sample corresponds to exactly one of finitely many classes. Examples of this are samples of the GM which correspond to one of two modes, or samples of MNIST which correspond to one of ten digits. The U-Turn then consists of starting with a sample from the data, run a backward diffusion model from time $t = 1$ to $t = t_0$, which noises the sample, and then run the forward diffusion model from time $t = t_0$ to $t = 1$ with noise independent from the backward run.

We are then interested in the probability that the sample before the backward and forward passes belongs to the same class as the sample after them. For $t_0 \approx 1$, this probability is close to 1. For $t_0 \approx 0$, this probability is close to the underlying probability of the diffusion model generating a sample of the given class. By running this for different $t_0$, we can find at what times it is decided to what class the samples belong to. Having found those times, our goal is to have a model that generates samples for each class according to the probabilities that they appear in the dataset. We can then improve the accuracy of the model on this by training on these times.

As a simple example, we train a U-Net (see Appendix D for details) to parameterize the Variance Preserving SDE from Song et al. (2021) to generate either the 0 or 1 digits from MNIST. The dataset we train on consists of 20% 1 digits and 80% 0 digits. We then measure how well is this model in generating samples that represent this asymmetry. The model is trained on approximately 7400 samples for 9 epochs, by sampling times in $[0, 1]$ uniformly as described in the beginning of this section. We then generate 18500 new samples running this model using 1000 discretization steps. [1] Among the 18500 generated samples, 88.2% are digits 0. (For determining this, we used a discriminator with 99.2% accuracy on MNIST, see Appendix D for details.)

We then test our proposed method. First, we determine at what time the digit that the sample represents is decided. We do this with the U-Turn method described above. Note that to do this, we use the model that we already trained. The results are in Figure 2. We find that the times important for deciding the digit are early in the generation for $t \in [0.2, 0.6]$ and mostly concentrated on $t \in [0.3, 0.5]$.

We now train from scratch a model on 7400 samples for 9 epochs as before, except that we do not sample the times uniformly. We instead sample times with probability $1/2$ uniformly in the interval $[0.3, 0.5]$ and with probability $1/2$ uniformly outside that interval. We then generate 18500 new samples with this new model using 1000 discretization steps, and find that 81.0% are 0s. We similarly consider sampling times with probability $1/2$ uniformly in the interval $[0.2, 0.6]$ and with probability $1/2$ outside that interval, generate samples, and find that 81.1% are 0s. This validates our hypothesis in the simple case of MNIST.

Although our theoretical analysis is for the probability flow ODE on the two-mode GM data distribution, this example on MNIST shows that the ideas developed here can be useful to the SDE generative models used in practice for real data.

---

[1]This amount of discretization steps is much larger than what is needed for MNIST, and we do it this way to make sure that the error is not coming from the integration of the SDE but from the training alone.

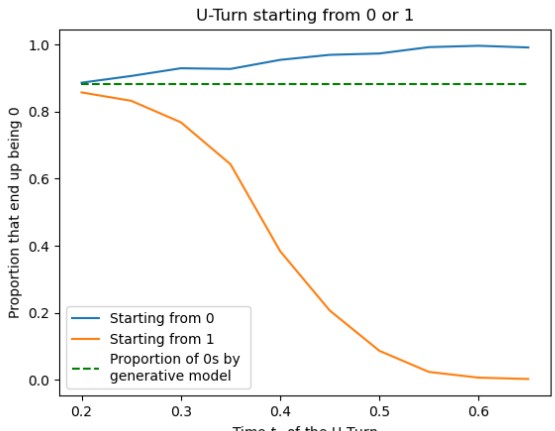

Figure 2: For $t_0 \in [0.2, 0.65]$, we plot the proportion of 0s that we get by doing the U-Turn at time $t_0$ starting from either 0 or 1 at time $t = 1$. On dashed green, we plot $y = .882$ which is the estimated proportion of 0s that the diffusion model generates starting from noise.

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
