## A  PROOF OF PROPOSITION 1

To prove Proposition 1, we will use the following three Lemmas that follow directly from Albergo et al. (2023) (Appendix A)

**Lemma 1.** *Let $a \sim \rho$ and $z \sim \mathcal{N}(0, Id_d)$. The law of the interpolant $I_t = \alpha_t z + \beta_t a$ coincides with the law of the solution of the probability flow ODE*

$$\dot{X}_t = \frac{\alpha_t \dot{\alpha}_t + \sigma^2 \beta_t \dot{\beta}_t}{\alpha_t^2 + \sigma^2 \beta_t^2} X_t + \frac{\alpha_t(\alpha_t \dot{\beta}_t - \dot{\alpha}_t \beta_t)}{\alpha_t^2 + \sigma^2 \beta_t^2} \mu \tanh\left(h + \frac{\beta_t \mu \cdot X_t}{\alpha_t^2 + \sigma^2 \beta_t^2}\right), \quad X_0 \sim \mathcal{N}(0, Id_d) \tag{15}$$

*where $h$ is such that $e^h/(e^h + e^{-h}) = p$.*

**Lemma 2.** *Let $a \sim p\mathcal{N}(\kappa, 1) + (1-p)\mathcal{N}(-\kappa, 1)$ and $z \sim \mathcal{N}(0, 1)$. The law of the interpolant $I_t = \alpha_t z + \beta_t a$ coincides with the law of the solution of the probability flow ODE. In*

$$\dot{X}_t = \frac{\alpha_t \dot{\alpha}_t + \beta_t \dot{\beta}_t}{\alpha_t^2 + \beta_t^2} X_t + \frac{\kappa \alpha_t(\alpha_t \dot{\beta}_t - \dot{\alpha}_t \beta_t)}{\alpha_t^2 + \beta_t^2} \tanh\left(h + \frac{\kappa \beta_t X_t}{\alpha_t^2 + \beta_t^2}\right), \quad X_0 \sim \mathcal{N}(0, 1) \tag{16}$$

*where $h$ is such that $e^h/(e^h + e^{-h}) = p$.*

**Lemma 3.** *Let $a \sim \mathcal{N}(\pm\mu, \sigma^2 Id_d)$ and $z \sim \mathcal{N}(0, Id_d)$. The law of the interpolant $I_t = \alpha_t z + \beta_t a$ coincides with the law of the solution of the probability flow ODE*

$$\dot{X}_t = \frac{\alpha_t \dot{\alpha}_t + \sigma^2 \beta_t \dot{\beta}_t}{\alpha_t^2 + \sigma^2 \beta_t^2} X_t \pm \frac{\alpha_t(\alpha_t \dot{\beta}_t - \dot{\alpha}_t \beta_t)}{\alpha_t^2 + \sigma^2 \beta_t^2} \mu, \quad X_0 \sim \mathcal{N}(0, Id_d). \tag{17}$$

*Proof of Proposition 1.* **First phase.** We have $\tau_t = \frac{\kappa t}{\sqrt{d}}$ since $t \in [0, 1]$. Plugging in $\alpha_t = 1 - \tau_t$ and $\beta_t = \tau_t$ into the velocity field from Lemma 1 yields

$$\dot{X}_t = \frac{\kappa}{\sqrt{d}}\left(-X_t + \mu \tanh\left(h + \kappa t \frac{\mu \cdot X_t}{\sqrt{d}}\right)\right) + O\left(\frac{1}{d}\right). \tag{18}$$

We then have, with $\nu_t = \mu \cdot X_t / \sqrt{d}$,

$$\dot{\nu}_t = \kappa \tanh\left(h + \kappa t \nu_t\right) + O\left(\frac{1}{\sqrt{d}}\right). \tag{19}$$

Taking $d \to \infty$ yields the limiting ODE for $\nu_t$. From Lemma 2, we get that this the 1-dimensional velocity field associated to the interpolant $I_t = \sqrt{1-t^2}z + ta$ that transports $z \sim \mathcal{N}(0, 1)$ at $t = 0$ to $a \sim p\mathcal{N}(\kappa, 1) + (1-p)\mathcal{N}(-\kappa, 1)$ at $t = 1$.

Let $X_t^\perp = X_t - \frac{\mu \cdot X_t}{d}\mu$. We have from equation 18

$$\dot{X}_t^\perp = -\frac{\kappa}{\sqrt{d}}X_t^\perp. \tag{20}$$

Since this is a linear ODE with initial condition Gaussian, we have

$$\dot{X}_t^\perp \sim \mathcal{N}\left(0, \sigma_t^2 Id_{d-1}\right). \tag{21}$$

Further, equation 20 gives $\dot{X}_t^\perp = O(1/\sqrt{d})$ meaning that for $t \in [0, 1]$

$$\lim_{d \to \infty} \sigma_t = 1. \tag{22}$$

**Second phase.** For $t \in [1, 2]$, we have $\tau_t = \left(1 - \frac{\kappa}{\sqrt{d}}\right)(2t - 1) + \frac{\kappa}{\sqrt{d}}$. Again using Lemma 1, we get

$$\dot{X}_t = \frac{-(2-t) + \sigma^2(t-1)}{(2-t)^2 + \sigma^2(t-1)^2} X_t + \frac{(2-t)\tanh\left(h + \frac{(t-1)\mu \cdot X_t + \kappa \frac{\mu \cdot X_t}{\sqrt{d}}}{(2-t)^2 + \sigma^2(t-1)^2}\right)}{(2-t)^2 + \sigma^2(t-1)^2}\mu + O\left(\frac{1}{\sqrt{d}}\right). \tag{23}$$

Writing $\nu_t = \frac{\mu \cdot X_t}{\sqrt{d}}$, this implies

$$\dot{\nu}_t = \frac{-(2-t) + \sigma^2(t-1)}{(2-t)^2 + \sigma^2(t-1)^2}\nu_t + \frac{(2-t)\sqrt{d}\tanh\left(h + \frac{(t-1)\sqrt{d}\nu_t + \kappa\nu_t}{(2-t)^2 + \sigma^2(t-1)^2}\right)}{(2-t)^2 + \sigma^2(t-1)^2} + O_d(1). \quad (24)$$

Let us calculate the initial condition for $\nu_t$ at $t = 1$. Write $a = sm + z$ where $p = \mathbb{P}(s = 1) = 1 - \mathbb{P}(s = -1)$ and $z \sim \mathcal{N}(0, \mathrm{Id}_d)$. Then

$$\frac{\mu \cdot I_{t=1}}{\sqrt{d}} \overset{(d)}{=} Z + \kappa s + O\left(\frac{1}{\sqrt{d}}\right)$$

where $Z \sim \mathcal{N}(0, 1)$. This means that for $\kappa$ large enough, then $|h| < \left|\frac{(t-1)\sqrt{d}\nu_t + \kappa\nu_t}{(2-t)^2 + \sigma^2(t-1)^2}\right|$ with high probability. This implies $\nu_t$ will not change sign during its trajectory, since whenever $\nu_t = o(\sqrt{d})$, the $\tanh$ term will dominate in equation 24. Hence, the following approximation is valid

$$\tanh\left(h + \frac{(2t-1)\sqrt{d}\nu_t + \kappa\nu_t}{1 + (\sigma^2 - 1)(2t-1)^2}\right) = \tanh\left(\sqrt{d}\nu_t\right) = \mathrm{sgn}(\nu_t) \quad (25)$$

We then use this approximation in the ODEs for $X_t$ to get

$$\dot{X}_t = \frac{-(2-t) + \sigma^2(t-1)}{(2-t)^2 + \sigma^2(t-1)^2}X_t + \frac{(2-t)\mathrm{sgn}(\nu_t)}{(2-t)^2 + \sigma^2(t-1)^2}\mu + O\left(\frac{1}{\sqrt{d}}\right). \quad (26)$$

Let $M_t = \mu \cdot X_t / d$. We get the induced equation

$$\dot{M}_t = \frac{-(2-t) + \sigma^2(t-1)}{(2-t)^2 + \sigma^2(t-1)^2}M_t + \frac{(2-t)\mathrm{sgn}(M_t)}{(2-t)^2 + \sigma^2(t-1)^2} + O\left(\frac{1}{\sqrt{d}}\right). \quad (27)$$

From the analysis of the first phase we had

$$\nu_1 \sim p\mathcal{N}(\kappa, 1) + (1-p)\mathcal{N}(-\kappa, 1). \quad (28)$$

We argued above that the sign of $\nu_t$ will be preserved for $t \in [1, 2]$ with probability going to 1 as $\kappa$ tends to $\infty$. This means that

$$M_2 = p^\kappa \delta_1 + (1 - p^\kappa)\delta_{-1} \quad (29)$$

where $p^\kappa$ is such that $\lim_{\kappa \to \infty} p^\kappa = p$.

As in the first phase, we let $X_t^\perp = X_t - \frac{\mu \cdot X_t}{d}\mu$. We have from equation 26 that

$$\dot{X}_t^\perp = \frac{-(2-t) + \sigma^2(t-1)}{(2-t)^2 + \sigma^2(t-1)^2}X_t^\perp + O\left(\frac{1}{\sqrt{d}}\right). \quad (30)$$

Since this is a linear ODE with initial condition Gaussian, we have

$$\dot{X}_t^\perp \sim \mathcal{N}\left(0, \sigma_t^2 \mathrm{Id}_{d-1}\right). \quad (31)$$

Under the change of variables $t(s) = s + 1$ we get that the ODE becomes

$$\dot{X}_s^\perp = \frac{-(1-s) + \sigma^2 s}{(1-s)^2 + \sigma^2 s^2}X_s^\perp. \quad (32)$$

By taking one coordinate $i \in \{1, \cdots, d-1\}$ of $X_s^\perp$ we get from Lemma 3 that this is the velocity field associated with the interpolant $I_s = \sqrt{1 - s^2}z + sa$ where $z \sim \mathcal{N}(0, 1)$ is transported to $a \sim \mathcal{N}(0, \sigma^2)$ as desired. $\qquad \square$

# B DERIVATIONS OF LEARNING RESULTS

## B.1 DERIVATION OF FIRST PHASE

Let $t \in [0, 1]$ so that

$$x_t^\mu = \left(1 - \frac{\kappa t}{\sqrt{d}}\right)x_0^\mu + \frac{\kappa t}{\sqrt{d}}x_1^\mu$$

Consider a denoiser parametrized as

$$f(x) = cx + u \tanh\left(b + \frac{w \cdot x}{\sqrt{d}}\right)$$

We introduce the following overlaps which we assume to be of order 1 in $d$

$$p_\eta^\mu = s^\mu \frac{z^\mu \cdot w}{d}, \quad \omega = \frac{\mu \cdot w}{d}, \quad r = \frac{\|w\|^2}{d} \quad q_\xi^\mu = s^\mu \frac{x_0^\mu \cdot u}{d}, \quad q_\eta^\mu = s^\mu \frac{z^\mu \cdot u}{d}, \quad m = \frac{\mu \cdot u}{d}, \quad q = \frac{\|u\|^2}{d}$$

We note that

$$\phi^\mu = f(x_t^\mu) = \tanh\left(b + \kappa t \sigma s^\mu p_\eta^\mu + \kappa t \omega s^\mu + \sqrt{r} Z^\mu + o_d(1)\right)$$

where $Z^\mu \sim \mathcal{N}(0,1)$. We now compute the loss

$$
\begin{aligned}
\frac{1}{d} \sum_\mu \|x_1^\mu - f(x_t^\mu)\|^2 &= \frac{1}{d} \sum_\mu \left\| x_1^\mu - c((1 - \kappa t/\sqrt{d})x_0^\mu + (\kappa t/\sqrt{d})x_1^\mu) - u\phi^\mu \right\|^2 \\
&= \frac{1}{d} \sum_\mu \|x_1^\mu - cx_0^\mu - u\phi^\mu\|^2 + o_d(1) \\
&= \sum_\mu 1 + \sigma^2 + c^2 + \frac{\|u\|^2}{d}(\phi^\mu)^2 - 2\left((\mu s^\mu + \sigma z^\mu)(1 - c\kappa t/\sqrt{d}) - cx_0^\mu\right) \cdot \frac{u}{d}\phi^\mu + o_d(1) \\
&= \sum_\mu 1 + \sigma^2 + c^2 + q(\phi^\mu)^2 - 2(m + \sigma q_\eta^\mu - cq_\xi^\mu)s^\mu \phi^\mu + o_d(1)
\end{aligned}
$$

We follow the same style of calculation as Cui et al. (2024) to compute the partition function. First we write the partition function

$$\mathcal{Z} = \int d\theta \, e^{-\frac{\gamma}{2} \hat{R}_t(\theta)}$$

$$= \int dc \, du \, dw \, db \, e^{-\frac{\gamma d}{2}\left(\sum_\mu 1 + \sigma^2 + c^2 + \frac{\|u\|^2}{d}(\phi^\mu)^2 - 2(\frac{u \cdot \mu}{d} + \sigma \frac{u \cdot z^\mu}{d} - c\frac{u \cdot x_0^\mu}{d})s^\mu \phi^\mu\right) - \frac{\gamma \lambda}{2}\|u\|^2 - \frac{\gamma \ell}{2}\|w\|^2}$$

Next we introduce overlaps into the integral and their conjugates by Dirac-Fourier, which we will denote as the vectors $\zeta$ and $\hat{\zeta}$ to simplify notation, and rearrange to integrate $u, w$

$$\mathcal{Z} = \int dc \, db \, d\zeta \, d\hat{\zeta} \, e^{d\left(\frac{1}{2}\hat{q}q + \hat{m}m + \frac{1}{2}\hat{r}r + \hat{\omega}\omega + \sum_{\mu=1}^n (q_\xi^\mu \hat{q}_\xi^\mu + q_\eta^\mu \hat{q}_\eta^\mu + p_\eta^\mu \hat{p}_\eta^\mu)\right) - \frac{\gamma}{2}\sum_\mu (1 + \sigma^2 + c^2 + q(\phi^\mu)^2 - 2(m + \sigma q_\eta^\mu - cq_\xi^\mu)s^\mu \phi^\mu)}$$

$$\int du \, e^{-\frac{\hat{q} + \gamma \lambda}{2}\|u\|^2 - u \cdot \left(\hat{m}\mu + \sum_{\mu=1}^n \hat{q}_\xi x_0^\mu + \hat{q}_\eta z^\mu\right)} \int dw \, e^{-\frac{\hat{r} + \gamma \ell}{2}\|w\|^2 - w \cdot \left(\hat{\omega}\mu + \sum_{\mu=1}^n \hat{p}_\eta z^\mu\right)}$$

Next we evaluate the $u, w$ integrals to get

$$e^{d\left(\frac{1}{2}\log(\hat{q}+\gamma\lambda) + \frac{1}{2}\log(\hat{r}+\gamma\ell) + \frac{1}{2(\hat{q}+\gamma\lambda)}\frac{1}{d}\|\hat{m}\mu + \sum_{\mu=1}^n \hat{q}_\xi^\mu x_0^\mu + \hat{q}_\eta^\mu z^\mu\|^2 + \frac{1}{2(\hat{r}+\gamma\ell)}\frac{1}{d}\|\hat{\omega}\mu + \sum_{\mu=1}^n \hat{p}_\eta^\mu z^\mu\|^2\right)}$$

$$= e^{d\left(\frac{1}{2}\log(\hat{q}+\gamma\lambda) + \frac{1}{2}\log(\hat{r}+\gamma\ell) + \frac{1}{2(\hat{q}+\gamma\lambda)}(\hat{m}^2 + \sum_{\mu=1}^n (\hat{q}_\xi^\mu)^2 + (\hat{q}_\eta^\mu)^2) + \frac{1}{2(\hat{r}+\gamma\ell)}(\hat{\omega}^2 + \sum_{\mu=1}^n (\hat{p}_\eta^\mu)^2) + O(1/\sqrt{d})\right)}$$

We now assume a sample-symmetry ansatz on the overlaps which means that $q_\xi^\mu = q_\xi$ for every $\mu$ are all equal, and the same for $\hat{q}_\xi^\mu, q_\eta^\mu, \hat{q}_\eta^\mu, p_\eta^\mu, \hat{p}_\eta^\mu$. We then take $d \to \infty$, rescale all conjugates with $\gamma$, change all conjugates signs except for $\hat{q}$ and $\hat{r}$ for cleaner equations, and take $\gamma \to \infty$. This gives us the following effective field (log partition function)

$$
\begin{aligned}
\log \mathcal{Z}(\mathcal{D}) = \text{extr}\Bigg\{ &- \frac{n}{2}\left(-2(\sigma q_\eta + m - cq_\xi)\overline{\phi s} + c^2 + q\overline{\phi^2}\right) \\
&+ \frac{q\hat{q}}{2} + \frac{r\hat{r}}{2} - m\hat{m} - \omega\hat{\omega} - n(q_\xi \hat{q}_\xi + q_\eta \hat{q}_\eta + p_\eta \hat{p}_\eta) \\
&+ \frac{\hat{m}^2 + n(\hat{q}_\xi^2 + \hat{q}_\eta^2)}{2(\lambda + \hat{q})} + \frac{\hat{\omega}^2 + n\hat{p}_\eta^2}{2(\ell + \hat{r})} \Bigg\}
\end{aligned}
$$

Taking gradients we get the following saddle-point equations

$$\begin{cases} (\sigma q_\eta + m - cq_\xi)\overline{\phi's} - q\overline{\phi\phi'} = 0 \\ c = -\overline{\phi}sq_\xi \\ p_\eta = \frac{\hat{p}_\eta}{\ell+\hat{r}} \\ \omega = \frac{\hat{\omega}}{\ell+\hat{r}} \\ r = \frac{\hat{\omega}^2+n\hat{p}_\eta^2}{(\ell+\hat{r})^2} = \omega^2 + np_\eta^2 \\ q_\xi = \frac{\hat{q}_\xi}{\lambda+\hat{q}} \\ q_\eta = \frac{\hat{q}_\eta}{\lambda+\hat{q}} \\ m = \frac{\hat{m}}{\lambda+\hat{q}} \\ q = \frac{\hat{m}^2+n(\hat{q}_\xi^2+\hat{q}_\eta^2)}{(\lambda+\hat{q})^2} = m^2 + nq_\xi^2 + nq_\eta^2 \end{cases} \qquad \begin{cases} \hat{p}_\eta = (\kappa t)((\sigma q_\eta + m - cq_\xi)\overline{\phi's} - q\overline{\phi'\phi}) = 0 \\ \hat{\omega} = (n\kappa t)((\sigma q_\eta + m - cq_\xi)\overline{\phi'} - q\overline{\phi'\phi s}) \\ \hat{r} = -n((\sigma q_\eta + m - cq_\xi)\overline{Z\phi's} - q\overline{Z\phi\phi'})/\sqrt{r} \\ \hat{q}_\xi = -c\overline{\phi}s \\ \hat{q}_\eta = \sigma\overline{\phi}s \\ \hat{m} = n\overline{\phi}s \\ \hat{q} = n\overline{\phi}^2 \end{cases}$$

Combining the equations for $c$ and $q_\xi$ we get that $c = c\overline{\phi s}^2/(\lambda + n\overline{\phi}^2)$. We now argue that $c = 0$ almost surely, since otherwise $\overline{\phi} = 0$ on a non-zero measure, implying $b = \omega = 0$, which then results in all overlaps being zero, giving a suboptimal log partition function. This can be seen more explicitly by noting that the log partition function is zero for $c \neq 0$, but for $c = 0$ it is instead

$$\log \mathcal{Z}(\mathcal{D}) = \text{extr}_{\omega,b} \left\{ n\frac{\overline{\phi s}^2(\sigma^2+n)}{2\left(\lambda + n\overline{\phi}^2\right)} - \frac{1}{2}\ell\omega^2 \right\}$$

which has positive values for example at $\omega = 0, b \neq 0$. The above formulation is also useful for solving for the overlaps numerically.

### B.1.1 ARGUMENT FOR COROLLARY 1

We now focus on $n \to \infty$ and on verifying that $\omega = \kappa t, b = \tanh^{-1}(\overline{s})$ is a solution. We will need the following preliminary claims.

**Lemma 4.** *Let* $\phi^\mu = \tanh(b + \kappa t\omega s^\mu + \omega Z^\mu)$. *If* $\omega = \kappa t$ *and* $b = \tanh^{-1}(\overline{s})$ *then* $\overline{\phi - s} = 0$

*Proof.* Let $\phi_\pm^\mu = \phi^\mu|_{s^\mu=\pm1}$ and $\phi_0 = \phi^\mu|_{s^\mu=0}$. Then

$$\overline{\phi - s} = \overline{p}\overline{(\phi_+ - 1)} + (1 - \overline{p})\overline{(\phi_- + 1)}$$
$$= \int dz\, e^{-\frac{z^2}{2}} \left\{ \overline{p}(\phi_+ - 1) + (1 - \overline{p})(\phi_- + 1) \right\}$$
$$= \int dz\, e^{-\frac{z^2+(\kappa t)^2}{2}} \left\{ e^{\kappa tz}\overline{p}(\phi_0 - 1) + e^{-\kappa tz}(1 - \overline{p})(\phi_0 + 1) \right\} \quad z \to z \mp \kappa t$$

Finally note that the integrad is zero for all $z$ if

$$\phi_0 = \frac{e^{\kappa tz}\overline{p} - e^{-\kappa tz}(1 - \overline{p})}{e^{\kappa tz}\overline{p} + (1 - \overline{p})e^{-\kappa tz}} = \tanh\left(\tanh^{-1}(\overline{s}) + \kappa tz\right)$$

which occurs for $\omega = \kappa t$ and $b = \tanh^{-1}(\overline{s})$. $\qquad\square$

**Corollary 7.** *Let* $\phi^\mu = \tanh(b + \kappa t\omega s^\mu + \omega Z^\mu)$. *If* $\omega = \kappa t$ *and* $b = \tanh^{-1}(\overline{s})$ *then for any function g where* $g(Z \pm \kappa t)(\phi \mp 1)$ *exist we have*

$$\overline{g(Z + s\kappa t)(\phi - s)} = 0$$

*In particular,*

$$\overline{\phi - s} = \overline{\phi(\phi - s)} = \overline{\phi'(\phi - s)} = \overline{(Z + s\kappa t)\phi'(\phi - s)} = 0$$

Solving for $q_\eta, m, q_\xi, q$ and plugging into the equation for $b$ we get

$$(\lambda + n\overline{\phi^2})\,\overline{\phi's} = n\overline{\phi s}\,\overline{\phi\phi'}$$

Taking $n \to \infty$, to leading order in $n$ the equality becomes $(\overline{\phi^2})(\overline{\phi's}) - (\overline{\phi'\phi})(\overline{\phi s}) = 0$ which holds by Corollary 7.

Using the independence of $s^\mu, Z^\mu$ and taking the limit of infinitely many $Z^\mu$, we can use Stein's lemma to rewrite the $\hat{r}$ equation as

$$\hat{r} = -n((\sigma q_\eta + m - cq_\xi)\overline{\phi''s} - q\overline{(\phi\phi')'})$$

Plugging in $q_\eta, m, q_\xi, q$ in gives

$$\hat{r}(\lambda + n\overline{\phi^2})^2 = -n((\lambda + n\overline{\phi^2})(\sigma^2 + n)\overline{\phi''s}\,\overline{\phi s} - n(\sigma^2 + n)\overline{\phi s}^2\,\overline{(\phi\phi')'})$$

Plugging $q_\eta, m, q_\xi, q$ into equations for $\omega$ and $\hat{\omega}$ gives

$$\omega(\ell + \hat{r})(\lambda + n\overline{\phi^2})^2 = (n\kappa t)((\lambda + n\overline{\phi^2})(\sigma^2 + n)(\overline{\phi'\,\phi s}) - n(\sigma^2 + n)\overline{\phi s}^2\overline{\phi'\phi s})$$

For large $n$ we get

$$-\omega((\overline{\phi^2})(\overline{\phi''s})(\overline{\phi s}) - (\overline{\phi s}^2)(\overline{(\phi\phi')'})) = (\kappa t)((\overline{\phi^2})(\overline{\phi'})(\overline{\phi s})) - (\overline{\phi s}^2)(\overline{\phi'\phi s}))$$

$$\omega = (\kappa t)\frac{((\overline{\phi'}) - (\overline{\phi'\phi s}))}{-((\overline{\phi''s}) - ((\overline{\phi\phi')'}))}$$

$$= (\kappa t)\frac{\overline{(\phi - \frac{1}{2}\phi^2 s)'}}{-\overline{(\phi - \frac{1}{2}\phi^2 s)''s}}$$

Finally note that

$$\overline{(\phi - \frac{1}{2}\phi^2 s)'} + \overline{(\phi - \frac{1}{2}\phi^2 s)''s} = -\frac{1}{2\kappa t}\overline{[(\phi - s)^2]'\,(s\kappa t + Z)} = 0$$

applying Corollary 7.

## B.2 DERIVATION OF SECOND PHASE

We now consider times $t \in [1, 2]$ which means we have

$$x_t^\mu = (2 - t)\left(1 - \frac{\kappa}{\sqrt{d}}\right)x_0^\mu + \left(\frac{\kappa}{\sqrt{d}} + \left(1 - \frac{\kappa}{\sqrt{d}}\right)(t - 1)\right)x_1^\mu.$$

We change variables to $\tau = t - 1$ and consider $\tau \in [0, 1]$ so that

$$x_\tau^\mu = (1 - \tau)\left(1 - \frac{\kappa}{\sqrt{d}}\right)x_0^\mu + \left(\frac{\kappa}{\sqrt{d}} + \left(1 - \frac{\kappa}{\sqrt{d}}\right)\tau\right)x_1^\mu.$$

We compute the loss for a single datapoint, defining $\nu^\mu = s^\mu\phi\left(w \cdot x_\tau^\mu + b\right)$

$$\frac{1}{d}\left\|x_1^\mu - c\left((1 - \tau)\left(1 - \frac{\kappa}{\sqrt{d}}\right)x_0^\mu + \left(\frac{\kappa}{\sqrt{d}} + \left(1 - \frac{\kappa}{\sqrt{d}}\right)\tau\right)x_1^\mu\right) - u\phi\left(w \cdot x_\tau^\mu + b\right)\right\|^2$$

$$= \frac{1}{d}\left\|x_1^\mu - c((1 - \tau)x_0^\mu + \tau x_1^\mu) - us^\mu\nu^\mu\right\|^2 + o_d(1)$$

$$= \frac{1}{d}\left\|(1 - c\tau)(\sigma z^\mu + s^\mu\mu) - c(1 - \tau)x_0^\mu - us^\mu\nu^\mu\right\|^2 + o_d(1)$$

$$= (1 + \sigma^2)(1 - c\tau)^2 + c^2(1 - \tau)^2 + \frac{\|u\|^2}{d} - \frac{2s^\mu\nu^\mu}{d}u \cdot ((1 - c\tau)(\sigma z^\mu + s^\mu\mu) - c(1 - \tau)x_0^\mu) + o_d(1)$$

$$= (1 + \sigma^2)(1 - c\tau)^2 + c^2(1 - \tau)^2 + q - 2\nu^\mu(1 - c\tau)(\sigma q_\eta^\mu + m) + 2\nu^\mu c(1 - \tau)q_\xi^\mu + o_d(1)$$

where we defined the overlaps

$$q = \frac{\|u\|^2}{d}, \quad q_\eta^\mu = s^\mu\frac{u \cdot z^\mu}{d}, \quad q_\xi^\mu = s^\mu\frac{u \cdot x_0^\mu}{d}, \quad m = \frac{u \cdot \mu}{d}$$

$$p_\eta^\mu = s^\mu \frac{w \cdot z^\mu}{d}, \quad p_\xi^\mu = s^\mu \frac{w \cdot x_0^\mu}{d}, \quad \omega = \frac{w \cdot \mu}{d}.$$

We also have

$$\nu^\mu = \phi\left(s^\mu w \cdot x_\tau^\mu + s^\mu b\right) = \tanh\left((1-\tau)s^\mu w \cdot x_0^\mu + \left(\tau + \frac{\kappa}{\sqrt{d}}\right)s^\mu w \cdot (\sigma z^\mu + s^\mu \mu) + s^\mu b\right)$$

$$= \tanh\left(d\left((1-\tau)p_\xi^\mu + \left(\tau + \frac{\kappa}{\sqrt{d}}\right)(\sigma p_\eta^\mu + \omega)\right) + s^\mu b\right)$$

$$\asymp \text{sign}\left(\sqrt{d}\left((1-\tau)p_\xi^\mu + \tau(\sigma p_\eta^\mu + \omega)\right) + \kappa(\sigma p_\eta^\mu + \omega)\right)$$

This gives the following

$$\log \mathcal{Z}(\mathcal{D}) = \text{extr}\Bigg\{-\frac{n}{2}\left((1+\sigma^2)(1-c\tau)^2 + c^2(1-\tau)^2 + q - 2(1-c\tau)(\sigma q_\eta + m)\overline{\nu} + 2c(1-\tau)q_\xi\overline{\nu}\right)$$

$$+ \frac{q\hat{q}}{2} - m\hat{m} - n(q_\xi\hat{q}_\xi + q_\eta\hat{q}_\eta) + \frac{\hat{m}^2 + n(\hat{q}_\xi^2 + \hat{q}_\eta^2)}{2(\lambda + \hat{q})}\Bigg\}$$

Taking gradients we get the following saddle-point equations

$$\begin{cases} q_\xi = \frac{\hat{q}_\xi}{\lambda + \hat{q}} = \frac{c(1-\tau)\overline{\nu}}{\lambda + n} \\ q_\eta = \frac{\hat{q}_\eta}{\lambda + \hat{q}} = \frac{(1-c\tau)\overline{\nu}}{\lambda + n} \\ m = \frac{\hat{m}}{\lambda + \hat{q}} = \frac{n(1-c\tau)\overline{\nu}}{\lambda + n} \\ q = \frac{\hat{m}^2 + n(\hat{q}_\xi^2 + \hat{q}_\eta^2)}{(\lambda + \hat{q})^2} = m^2 + nq_\xi^2 + n\sigma^2 q_\eta^2 \\ c = \frac{(1+\sigma^2)\tau - \tau(\sigma q_\eta + m)\overline{\nu} - (1-\tau)q_\xi\overline{\nu}}{(1-\tau)^2 + (1+\sigma^2)\tau^2} \end{cases} \qquad \begin{cases} \hat{q}_\xi = c(1-\tau)\overline{\nu} \\ \hat{q}_\eta = \sigma(1-c\tau)\overline{\nu} \\ \hat{m} = n(1-c\tau)\overline{\nu} \\ \hat{q} = n \end{cases}$$

$$c = \frac{(1+\sigma^2)\tau(\lambda + n) - \overline{\nu}^2\tau(\sigma + n)}{(\lambda + n)((1-\tau^2) + (1+\sigma^2)\tau^2) + \overline{\nu}^2\left((1-\tau)^2 - \tau^2(\sigma + n)\right)}$$

Corollary 3 simply follows from taking the $n \to \infty$ limit of this equations.

Lastly, we now argue that we can take $\overline{\nu} = 1$ without loss of generality. If we assume a sample symmetric ansatz for $p_\eta^\mu = p_\eta, p_\xi^\mu = p_\xi$, then $\overline{\nu}$ can either be $\pm 1$ depending on the sign of argument. Noting that $q, c$ are unchanged while $q_\eta, q_\xi, m, \hat{q}_\eta, \hat{q}_\xi, \hat{m}$ flip sign, we can conclude that the log partition function is invariant so $\overline{\nu} = 1$.

The characterizations of the learned parameters can be used to evaluate the MSE as a function of $t$, which we now describe, in the limit of $d \to \infty$ and then $n \to \infty$. For the first and second phase we obtain

$$\text{mse}_{\text{train}} = \text{mse}_{\text{test}} = \begin{cases} \sigma^2 + (1 - \overline{\phi^2}) & t \in [0,1] \\ \sigma^2(1-c\tau)^2 + c^2(1-\tau)^2 & t \in [1,2] \end{cases}$$

At $t = 0$, $\phi = \tanh(b) = 2(p - 1/2)$ hence the MSE is $\sigma^2 + 4p(1-p)$. At $t = 1$ we have $c = 0$ hence the MSE is $\sigma^2$, while at $t = 2$ we have $c = 1$ so the MSE is 0.

## C  ARGUMENTS FOR GENERATION

Combining equations 4 and 8 gives the exact velocity field

$$b_t(x) = \left(\dot{\beta}_t - \frac{\dot{\alpha}_t}{\alpha_t}\beta_t\right)\left(\frac{\beta_t\sigma^2}{\alpha_t^2 + \sigma^2\beta_t^2}x + \frac{\alpha_t^2}{\alpha_t^2 + \sigma^2\beta_t^2}\mu\tanh\left(\frac{\beta_t}{\alpha_t^2 + \sigma^2\beta_t^2}\mu \cdot x + h\right)\right) + \frac{\dot{\alpha}_t}{\alpha_t}x.$$
$$\tag{33}$$

where $\alpha_t = 1 - \tau_t$ and $\beta_t = \tau_t$ with $\tau_t$ from equation 12. Let $\hat{\theta}_t$ denote any overlap from the first phase (see equation 13) in the limit of $d \to \infty$ but for finite $n$, where $\theta_t$ denotes the corresponding overlap with $d \to \infty$ and then $n \to \infty$. From Results 1 and 2 and their Corollaries 1 and 3, we have that $|\hat{\theta}_t - \theta_t| = O_n(1/n)$ for all overlaps.

Since $X_t - \hat{X}_t$ is contained in $\mathrm{span}(u_t, \eta)$ which is in turn contained in $\mathrm{span}(\mu, \eta, \xi)$, it suffices to show that, after dividing by $d$, the projections of $X_t - \hat{X}_t$ onto $\mu$, $\eta$, and $\xi$ are $O(1/n)$ to show that $\frac{1}{d}\|X_t - \hat{X}_t\|$ is $O(1/n)$.

## C.1  ARGUMENT FOR RESULT 3

First, we note that as described in the paragraph above the statement of Result 3, we have that since in the first phase $q = m^2 + nq_\eta^2$ from Result 1 we get for $t \in [0, 1]$

$$\lim_{d \to \infty} \frac{\|u_t\|^2}{d} = \lim_{d \to \infty} \left(\frac{\mu \cdot u_t}{d}\right)^2 + \left(\frac{\eta \cdot u_t}{d}\right)^2$$

also since $q = m^2 + nq_\xi^2 + nq_\eta^2$ in the second phase, we get that for $t \in [1, 2]$

$$\lim_{d \to \infty} \frac{\|u_t\|^2}{d} = \lim_{d \to \infty} \left(\frac{\mu \cdot u_t}{d}\right)^2 + \left(\frac{\eta \cdot u_t}{d}\right)^2 + \left(\frac{\xi \cdot u_t}{d}\right)^2$$

where $\eta = \sigma \sum_{\mu=1}^{n} z^\mu$ and $\xi = \sum_\mu s^\mu x_0^\mu$ which implies that for any $w \in \mathrm{span}(\mu, \eta, \xi)^\perp$ with $\|w\|_2 = 1$ we have

$$\lim_{d \to \infty} \frac{w \cdot (X_2 - \hat{X}_2)}{\sqrt{d}} = 0.$$

### C.1.1  FIRST PHASE

We focus on $t \in [0, 1]$ and define

$$\epsilon_t^m = \frac{1}{\sqrt{d}}\mu \cdot (X_t - \hat{X}_t), \quad \epsilon_t^\eta = \frac{1}{\sigma^2 n\sqrt{d}}\eta \cdot (X_t - \hat{X}_t),$$

$$\delta_t = \frac{\beta_t}{\alpha_t^2 + \sigma^2\beta_t^2}, \quad \gamma_t = \frac{\alpha_t^2}{\alpha_t^2 + \sigma^2\beta_t^2},$$

$$M_t = \frac{\mu \cdot X_t}{\sqrt{d}}, \quad Q_t^\eta = \frac{\eta \cdot X_t}{\sigma^2 n\sqrt{d}}.$$

We have

$$\dot{\epsilon}_t^m = \frac{1}{\sqrt{d}}\mu \cdot (\dot{X}_t - \dot{\hat{X}}_t)$$

$$= \frac{1}{\sqrt{d}}\mu \cdot (b_t(X_t) - \hat{b}_t(\hat{X}_t))$$

$$= \frac{1}{\sqrt{d}}\mu \cdot \left(\dot{\beta} - \frac{\dot{\alpha}}{\alpha}\beta\right)\left(\sigma^2\delta_t(X_t - \hat{X}_t) + (c_t - \sigma^2\delta_t)X_t + (\gamma_t\mu - u_t)\tanh(\delta_t\mu \cdot X_t + h)\right.$$

$$\left. + u_t\left(\tanh(\delta_t\mu \cdot X_t + h) - \tanh(w_t \cdot X_t + b_t))\right) + \frac{\dot{\alpha}}{\alpha}\frac{1}{\sqrt{d}}\mu \cdot \left(X_t - \hat{X}_t\right)\right.$$

$$= \left(\dot{\beta} - \frac{\dot{\alpha}}{\alpha}\beta\right)\left(\sigma^2\delta_t\epsilon_t^m + (c_t - \sigma^2\delta_t)M_t + \sqrt{d}(\gamma_t - m_t)\tanh(\delta_t\mu \cdot X_t + h)\right.$$

$$\left. + \sqrt{d}m_t\left(\tanh(\delta_t\mu \cdot X_t + h) - \tanh(w_t \cdot \hat{X}_t/\sqrt{d} + b_t))\right)\right) + \frac{\dot{\alpha}}{\alpha}\epsilon_t^m$$

$$= \kappa\left(1 - t\frac{\dot{\alpha}}{\alpha}\right)\left(\frac{\delta_t}{\sqrt{d}}\epsilon_t^m + (c_t - \sigma^2\delta_t)\frac{M_t}{\sqrt{d}} + (\gamma_t - m_t)\tanh(\delta_t\mu \cdot X_t + h)\right.$$

$$\left. + m_t\left(\tanh(\delta_t\mu \cdot X_t + h) - \tanh(w_t \cdot \hat{X}_t/\sqrt{d} + b_t))\right)\right) + \frac{\dot{\alpha}}{\alpha}\epsilon_t^m$$

We now focus on the $\tanh$

$$\left| \tanh(\delta_t \mu \cdot X_t + h) - \tanh(w_t \cdot \hat{X}_t / \sqrt{d} + b_t) \right|$$

$$\leq \left| \left( \delta_t \mu - \frac{w_t}{\sqrt{d}} \right) \cdot X_t \right| + \left| \frac{w_t}{\sqrt{d}} \cdot (X_t - \hat{X}_t) \right| + |h - b_t|$$

$$\leq \left| \left( \delta_t \mu - \frac{w_t}{\sqrt{d}} \right) \left( \frac{\mu \mu^T}{d} + \frac{\eta \eta^T}{\sigma^2 n^2 d} \right) X_t \right| + \left| \frac{w_t}{\sqrt{d}} \left( \frac{\mu \mu^T}{d} + \frac{\eta \eta^T}{\sigma^2 n^2 d} \right) (X_t - \hat{X}_t) \right| + |h - b_t|$$

$$\leq \left| \left( \sqrt{d} \delta_t - \omega_t \right) M_t \right| + \left| (\delta_t Z - p_t^\eta) Q_t \right| + |\omega_t \epsilon_t^m| + |p_t^\eta \epsilon_t^\eta| + |h - b_t|$$

$$\leq \omega_t |\epsilon_t^m| + O\left( \frac{1}{n} \right) + O\left( \frac{1}{\sqrt{d}} \right).$$

Coming back to the ODE for $\dot{\epsilon}_t^m$, we get with high probability

$$|\dot{\epsilon}_t^m| = \kappa \gamma_t \omega_t |\epsilon_t^m| + O\left( \frac{1}{n} \right) + O\left( \frac{1}{\sqrt{d}} \right).$$

Since $\kappa \gamma_t \omega_t = \Theta_{n,d}(1)$ for $t \in [0, 1]$, we get that with high probability

$$\epsilon_{t=1}^m = O\left( \frac{1}{n} \right) + O\left( \frac{1}{\sqrt{d}} \right).$$

By performing a similar computation for the ODE for $\epsilon_t^\eta$, we get that with high probability

$$\epsilon_{t=1}^\eta = O\left( \frac{1}{n} \right) + O\left( \frac{1}{\sqrt{d}} \right).$$

### C.1.2   SECOND PHASE

We now turn to $t \in [1, 2]$ and define

$$\zeta_t^m = \frac{1}{d} \mu \cdot (X_t - \hat{X}_t), \quad \zeta_t^\eta = \frac{1}{\sigma^2 nd} \eta \cdot (X_t - \hat{X}_t) \quad \zeta_t^\xi = \frac{1}{nd} \xi \cdot (X_t - \hat{X}_t).$$

With high probability, we have the following ODEs hold

$$\frac{d}{dt} \zeta^m = \left( \beta(t) c_t + \frac{\dot{\alpha}(t)}{\alpha(t)} (1 - c_t \beta(t)) \right) \zeta^m + O\left( \frac{1}{n} \right), \tag{34}$$

$$\frac{d}{dt} \zeta^\eta = \left( \beta(t) c_t + \frac{\dot{\alpha}(t)}{\alpha(t)} (1 - c_t \beta(t)) \right) \zeta^\eta + O\left( \frac{1}{n} \right), \tag{35}$$

$$\frac{d}{dt} \zeta^\xi = \left( \beta(t) c_t + \frac{\dot{\alpha}(t)}{\alpha(t)} (1 - c_t \beta(t)) \right) \zeta^\xi + O\left( \frac{1}{n} \right). \tag{36}$$

from the initial condition $\zeta_1^m, \zeta_1^\eta = O(\frac{1}{\sqrt{d}}) + O(\frac{1}{n}), \zeta_1^\xi = 0$. This yields

$$\zeta_2^m, \zeta_2^\eta, \zeta_2^\xi = O(\frac{1}{n}) + O(\frac{1}{\sqrt{d}})$$

### C.2   ARGUMENT FOR COROLLARY 6

By Proposition 1, we know that

$$\lim_{\kappa \to \infty} \lim_{d \to \infty} \frac{\mu \cdot X_2}{d} \sim p \delta_1 + (1 - p) \delta_{-1}.$$

By Result 3, we get that

$$\lim_{n \to \infty} \lim_{d \to \infty} \frac{\mu \cdot (\hat{X}_2 - X_2)}{d} = 0.$$

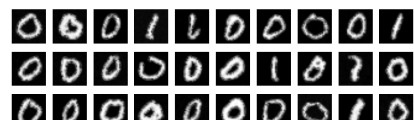
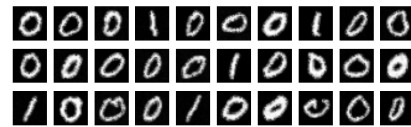

(a) Training times with
prob. $1/2$ on $[.2, .6]$

(b) Training times with
prob. $1/2$ on $[.3, .5]$

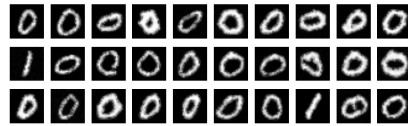

(c) Training times
uniform on $[0, 1]$

Figure 3: Non-cherry-picked samples from the three generative models considered. **(a)** Samples from the VP SDE, where the times for training are drawn with probability $1/2$ uniformly from $[.2, .6]$ and with probability $1/2$ uniformly outside. **(b)** Same as left panel except that with probability $1/2$ training times are sampled from $[.3, .5]$. **(c)** Samples from the VP SDE with training times that are uniform in $[0, 1]$.

Combining the last two equations gives the first claim from the Corollary.

Fix $w \perp \mu$, $\|w\| = 1$. Again by Proposition 1, we have that

$$\lim_{d \to \infty} \frac{w \cdot X_2}{\sqrt{d}} \sim \mathcal{N}(0, \sigma^2).$$

Also, Result 3 gives that

$$\lim_{n \to \infty} \lim_{d \to \infty} \frac{w \cdot (\hat{X}_2 - X_2)}{\sqrt{d}} = 0.$$

which combined with the previous equation gives the second claim from the Corollary.

## D  EXPERIMENTAL DETAILS

The model used for the MNIST experiment consists of a U-Net architecture (Ronneberger et al. (2015)), consisting of four downsampling and four upsampling blocks with two layers per block and output channels of 128, 128, 256, and 512, respectively. Attention mechanisms are integrated into the third downsampling block and the second upsampling block to enhance feature representation at multiple scales. The training of the denoiser is described in the main text. We then use this denoiser to estimate the score and run the Variance Preserving SDE (see equation (11) in Song et al. (2021).)

For the discriminative model, we use the MNIST digit classification model by Knight (2022) available on Hugging Face which achieves an accuracy of $99.1\%$ on MNIST classification.

As a sanity check, we show non-cherry-picked samples generated by the three models we considered in Figure 3.

## E  GENERAL TIME DILATION FORMULA

In this section, we generalize the time dilation formula from equation 12 for a Gaussian mixture with more than two modes. Although the arguments in Results 1 and 2 only hold for the two-mode GM, the fact that a more general time dilation formula exists suggests that these results could be extended to the GM with more than two modes.

Consider $\mu = \sum_{i=1}^m p_i \mathcal{N}\left(r_i, \sigma^2 I\right)$ where $r_i \in \mathbb{R}^d$ and $|r_i|$ goes to infinity with $d$, but $m, p_i, \sigma^2$ are constant with respect to $d$. If $X_t$ is the generative model associated with the interpolant $I_t = (1-t)z + ta$ where $z \sim \mathcal{N}(0, I)$ and $a \sim \mu$ (as we do in equation equation 3 in the main text) then $X_t$ estimates $p_i$ at times of the order $1/|r_i|$. We show this in Proposition 2 below by arguing that it is only at times of order $1/|r_i|$ that the denoiser associated to $r_i \cdot X_t/|r_i|$ is nontrivial. Hence, to estimate $p_i$ we require a time dilation $\tau_t$ such that there exists $a$ and $b$ with $b - a = \Theta_d(1)$ where

$$\tau_t = \Theta_d\left(\frac{1}{|r_i|}\right) \quad \text{for} \quad t \in [a, b]. \tag{37}$$

We specify next a time dilation that for every $i$ would ensure that Equation 37 is fulfilled. Assume $|r_1| \le |r_2| \le \cdots \le |r_m|$, let $n = m+1$ and let $\kappa > 0$. Then

$$\tau_t = \begin{cases} \frac{\kappa n t}{|r_m|} & \text{if } t \in [0, 1/n] \\ \frac{\kappa(nt-1)}{|r_{m-1}|} + \frac{\kappa}{|r_m|} & \text{if } t \in [1/n, 2/n] \\ \cdots & \\ \frac{\kappa(nt-(m-1))}{|r_1|} + \kappa\left(\frac{1}{|r_2|} + \cdots + \frac{1}{|r_m|}\right) & \text{if } t \in [(m-1)/n, m/n] \\ \left(1 - \kappa\left(\frac{1}{|r_1|} + \cdots + \frac{1}{|r_m|}\right)\right)t + \kappa\left(\frac{1}{|r_1|} + \cdots + \frac{1}{|r_m|}\right) & \text{if } t \in [m/n, 1] \end{cases} \tag{38}$$

Then we have that $p_i$ is learned when $t \in [(m-i)/n, (m-i+1)/n]$ and the $\sigma^2$ will be learned when $t \in [m/n, 1]$, giving rise to $m+1$ different phases. In the special case of $|r_i| = |r_{i+1}|$, both $p_i$ and $p_{i+1}$ will already be learned in $[(m-i)/n, (m-i+1)/n]$ so that the phase on the interval $[(m-i+1)/n, (m-i+2)/n]$ is unnecessary. Taking this consideration into account when using the general formula in equation 38 for the two-mode GM gives the time dilation formula from equation 12. The only difference is that the time dilation here maps $[0, 1]$ to $[0, 1]$ and the one in equation 12 maps $[0, 1]$ to $[0, 2]$.

**Proposition 2.** *Let $\mu = \sum_{i=1}^m p_i \mathcal{N}\left(r_i, \sigma^2 I\right)$ where $r_i \in \mathbb{R}^d$ and $|r_i| = \omega_d(1)$. Consider the interpolant $I_t = (1-t)z + ta$ where $z \sim \mathcal{N}(0, Id)$ and $a \sim \mu$. Let $X_t$ be the generative model associated to $I_t$ as in equation equation 3. Then $X_t$ learns the $p_i$ at times $\Theta_d\left(1/|r_i|\right)$.*

*Proof.* Fix $i$. Let $m_t = r_i \cdot I_t/|r_i|$. We have $m_t \overset{d}{=} (1-t)Z + t|r_i|m$ where $Z \sim \mathcal{N}(0, 1)$ and $m = r_i \cdot a/|r_i|^2 = \Theta_d(1)$. Let $\nu_t = r_i \cdot X_t/|r_i|$. By Lemma 5, $\nu_t$ obeys the self-consistent ODE

$$\dot{\nu}_t = \frac{\nu_t}{t} - \frac{\eta_t(\nu_t)}{t} \tag{39}$$

where $\eta_t(\nu)$ is the denoiser for $\nu_t$

$$\eta_t(\nu) = \mathbb{E}[Z|m_t = \nu] = \mathbb{E}[Z|(1-t)Z + t|r_i|m = \nu].$$

By Lemma 6, since $|r_i| = \omega_d(1)$, the only times where this denoiser is nontrivial are $t = \Theta_d\left(1/|r_i|\right)$. We note that to estimate $p_i$ we need to estimate $\nu_t$, which requires spending a constant length of time in the nontrivial times of the ODE in equation 39, which are the nontrivial times for the denoiser. Indeed, $p_i$ is learned on that interval, and if the length of that interval goes to 0 as $d$ goes to infinity, we cannot estimate $p_i$. $\qquad \square$

**Lemma 5.** *Let $\mu = \sum_{i=1}^m p_i \mathcal{N}\left(r_i, \sigma^2 I\right)$ where $r_i \in \mathbb{R}^d$. Consider the interpolant $I_t = (1-t)z + ta$ where $z \sim \mathcal{N}(0, Id)$ and $a \sim \mu$. Let $X_t$ be the generative model associated to $I_t$ from Lemma 5. Fix $i$ and let $m_t = r_i \cdot I_t/|r_i|$ and $\nu_t = r_i \cdot X_t/|r_i|$. Then with $\eta_t(\nu) = \mathbb{E}[Z|m_t = \nu]$ we have*

$$\dot{\nu}_t = \frac{\nu_t}{t} - \frac{\eta_t(\nu_t)}{t}$$

*Proof.* We have from Appendix A, Albergo et al. (2023) that the velocity field $b_t(x)$ associated with $I_t = (1-t)z + ta$ where $a \sim \sum_{i=1}^m p_i \mathcal{N}(r_i, \mathrm{Id})$ can be written explicitly as

$$\frac{\sum_{i=1}^m p_i \left(r_i + \frac{\dot{c}_t}{2c_t}(x - tr_i)\right) \mathcal{N}(x \mid tr_i, c_t\mathrm{I})}{\sum_{i=1}^m p_i \mathcal{N}(x \mid tr_i, c_t\mathrm{I})} = \frac{\sum_{i=1}^m p_i \left(r_i + \frac{\dot{c}_t}{2c_t}(x - tr_i)\right) e^{\frac{2tr_i \cdot x - t^2|r_i|^2}{2((1-t)^2 + t^2)}}}{\sum_{i=1}^m p_i e^{\frac{2tr_i \cdot x - t^2|r_i|^2}{2((1-t)^2 + t^2)}}}$$

$$= \frac{\dot{c}_t}{2c_t} x + \frac{\sum_{i=1}^m p_i \left(1 - \frac{\dot{c}_t}{2c_t}t\right) e^{\frac{2tr_i \cdot x - t^2|r_i|^2}{2((1-t)^2 + t^2)}}}{\sum_{i=1}^m p_i e^{\frac{2tr_i \cdot x - t^2|r_i|^2}{2((1-t)^2 + t^2)}}} r_i$$

where $c_t = (1-t)^2 + t^2$. The denoiser $\eta_t(x) = \mathbb{E}[z|I_t = x]$ is

$$\eta_t(x) = x - tb_t(x) = \left(1 - \frac{\dot{c}_t}{2c_t}\right)x - \frac{\sum_{i=1}^m p_i \left(t - \frac{\dot{c}_t}{2c_t}t^2\right) e^{\frac{2tr_i \cdot x - t^2|r_i|^2}{2((1-t)^2 + t^2)}}}{\sum_{i=1}^m p_i e^{\frac{2tr_i \cdot x - t^2|r_i|^2}{2((1-t)^2 + t^2)}}} r_i. \qquad (40)$$

Fix $i$ and let $m_t = r_i \cdot I_t/|r_i|$ and $\nu_t = r_i \cdot X_t/|r_i|$. Since $\dot{X}_t = b(X_t)$, we get that

$$\dot{\nu}_t = \frac{r_i \cdot b(X_t)}{|r_i|} = \frac{\nu_t}{t} - \frac{1}{t}\frac{r_i \cdot \eta_t(X_t)}{|r_i|} = \frac{\nu_t}{t} - \frac{\eta_t(\nu_t)}{t},$$

where the denoiser for the $\nu_t$ is defined as $\eta_t(\nu) = \mathbb{E}[Z|m_t = \nu]$. The last step in the displayed equality follows since from equation 40 we get that $r_i \cdot \eta_t(x)/|r_i|$ depends on $x$ only through $\nu_t$. $\qquad\square$

**Lemma 6.** *Let $Z \sim \mathcal{N}(0,1)$ and $M \sim \mu$. Then for fixed $\gamma > 0$ we have that as $d \to \infty$*

$$\mathbb{E}[Z|Z + d^{-\gamma}M = x] \to x$$
$$\mathbb{E}[Z|Z + d^{\gamma}M = x] \to \mathbb{E}[Z] = 0$$

*Proof.* Let $f_{Z,X}(z,x)$ be the joint density of $Z$ and $X = Z + d^{-\gamma}M$ and $f_{Z,M}(z,m)$ the joint density of $Z$ and $M$. We note that $f_{Z,X}(z,x) = f_{Z,M}(z, d^\gamma(x - z)) = f_Z(z)f_M(d^\gamma(x - z))$

$$\mathbb{E}[Z|Z + d^{-\gamma}M = x] = \frac{\int z f_{Z,X}(z,x)dz}{\int f_{Z,X}(z,x)dz}$$

$$= \frac{\int z f_Z(z) f_M(d^\gamma(x - z))dz}{\int f_Z(z) f_M(d^\gamma(x - z))dz}$$

$$= \frac{\int z f_Z(z) d^\gamma f_M(d^\gamma(x - z))dz}{\int f_Z(z) d^\gamma f_M(d^\gamma(x - z))dz}$$

$$\to x$$

where the last step follows since $d^\gamma f_M(d^\gamma z)$ is an approximation to the identity. The other limit follows similarly. $\qquad\square$