# OpenReview forum: "Phase-aware Training Schedule Simplifies Learning in Flow-Based Generative Models"
_ICLR.cc/2025/Conference — Submitted to ICLR 2025_

### Official Review · Reviewer_19x1 · 2024-10-28

**Soundness:** 3
**Presentation:** 2
**Contribution:** 2
**Rating:** 5
**Confidence:** 4

**Summary:**

This paper deals with an important issue with flow-based and diffusion models, namely that the generative process is highly non-stationary, with certain phases of diffusion being almost 'trivial' and with short 'critical' time windows that have an overwhelming importance in the final generation. These decision points are also known as spontaneous symmetry breaking points or speciation points and are strongly connected with the theory of critical phase transitions in statistical physics (Raya, 2023; Ambrogioni, 2023; Biroli, 2024; Li, 2024).

The paper offers a thorough theoretical analysis of flow-based generative models under a mixture of Gaussian data in high dimension, which results in a time-dilation formula that optimizes the training process by a non-linear transformation of the time variable. The formula is designed to 'push' the symmetry breaking point that divide two predefined classes at infinity, which results in a more homogeneous training and in more balanced generation.

While the formulas have been derived for mixture of Gaussian data, the author also provides experiments that suggest their usefulness in real image datasets.

**Strengths:**

To my knowledge, this is one of the first papers to deal with this important problem. Most existing work on time and noise scheduling are only based on features of the forward process or on empirical performance by trial-and-error. Given the insights from statistical physics, the time is ripe for a more principled approach to noise scheduling and I highly appreciate the effort of the authors in providing a principled solution.

The paper is generally well-written and its results are relatively easy to read. while I did not check the details of the proofs, the theoretical analysis appears to be rigorous and well-motivated. The basic idea of removing the trivial phase of diffusion by changing the time axis is well-principled and it seems to lead to performance gains in simple models and to promising results in real image datasets.

**Weaknesses:**

1)
The paper misses some very important references on phase transitions in generative diffusion. The cited analysis of speciation times in Biroli (2024) was partially based on prior work on spontaneous symmetry breaking (Raya, 2023), which should be properly discussed. In fact, this was the first work to characterize symmetry-breaking phenomena as a function of the time variable and to suggest the separation in qualitatively different generative phases, which is fundamental to the approach the authors are proposing. The authors should also discuss the further developments in (Ambrogioni, 2023), and the more mathematical related work in (Li, 2024). Note that, while these results are not stated in terms of stochastic interpolants, they all translate directly to the setting considered in this submission.

2) The time dilation formula in Eq.10 can only be calibrated on a single symmetry-breaking point. This means that the proposed method can only improve the probabilistic calibration of a specific class separation. While I do think that this is a valid starting point, it would be more useful to have a formula that can recalibrate the sampling of data with multiple decision points and a more complex class structure.

3) The experiments on real image datasets are rather weak for the standard of a top international conference. It would be useful to see the analysis repeated on other datasets such as Cifar10, CelebA and ImageNet and on other class divisions. It would also be useful to compare the results with other noise scheduling methods commonly used in the literature.

References:
I) Raya, Gabriel, and Luca Ambrogioni. "Spontaneous Symmetry Breaking in Generative Diffusion Models." arXiv preprint arXiv:2305.19693 (2023).
II) Ambrogioni, Luca. "The statistical thermodynamics of generative diffusion models." arXiv preprint arXiv:2310.17467 (2023).
III) Li, Marvin, and Sitan Chen. "Critical windows: non-asymptotic theory for feature emergence in diffusion models." arXiv preprint arXiv:2403.01633 (2024).

**Questions:**

1) Could you connect your result to the theory of spontaneous symmetry breaking in generative diffusion models?

2) Could you offer some suggestions on how to extend your formula to the case with multiple classes that separate at different critical times?

---

> ### Author Response · Authors · 2024-11-24
>
> Thank you for your thorough review. We address your comments as follows.
>
> **W1. The paper misses some very important references on phase transitions in generative diffusion.** In the revised version of the paper, we now refer and discuss the relations with the work of Raya and Ambrogioni 2023, Ambrogioni 2023, and Li and Chen 2024. Please let us know if this addresses your concern.
>
> **Q1. Could you connect your result to the theory of spontaneous symmetry breaking in generative diffusion models?** Our results can be stated as follows in terms of symmetry breaking. (1) The results from Biroli et al (2024) translated to flow-based generative models show that for generating samples from the two-mode GM in dimension $d$, there is a spontaneous symmetry breaking at times of order $1/\sqrt{d}.$ (2) We then show in Proposition 1 that the critical window where the symmetry breaking happens can be made of constant length in $d$ through a time dilation, which we provide in Eq. (10) in the original paper. (3) Finally, analyzing the learning problem using this time dilation shows that the generative model can resolve the spontaneous symmetry breaking, stated formally in Results 1 and 2, and Corollary 5.
>
> **W2/Q2. The time dilation formula in Eq.10 can only be calibrated on a single symmetry-breaking point. ... It would be more useful to have a formula that can recalibrate the sampling of data with multiple decision points.** The time dilation formula in Eq. 10 can be extended to more than two modes as follows. Consider $\mu=\sum_{i=1}^{m}p_{i}\mathcal{N}\left(r_{i},\sigma^{2}\text{I}\right)$ where $r_{i}\in\mathbb{R}^{d}$ and $|r_{i}|$ goes to infinity with $d,$ but $m,p_{i},\sigma^{2}$ are constant with respect to $d.$ If $X_{t}$ is the generative model associated with the interpolant $I_{t}=(1-t)z+ta$ where $z\sim\mathcal{N}(0,\text{I})$ and $a\sim\mu,$ then $X_{t}$ estimates $p_{i}$ at times of the order $1/|r_{i}|.$ We show this in Proposition 2 in the Appendix, Section E, of the revised version of the paper by arguing that it is only at times of order $1/|r_{i}|$ that the denoiser associated to $r_{i}\cdot X_{t}/|r_{i}|$ is nontrivial. Hence, to estimate $p_{i}$ we require a time dilation $\tau_{t}$ such that there exists $a$ and $b$ with $b-a=\Theta_{d}(1)$ where
> $$
> \tau_{t} =\Theta_{d}\left(\frac{1}{|r_{i}|}\right) \text{  for  }t\in[a,b].
> $$
>
> From here we can derive the more general dilation formula that you asked for. We need to specify a dilation that for every $i$ ensures that the condition on $\tau_t$ is fulfilled. Assume $|r_{1}|\leq|r_{2}|\leq\cdots\leq|r_{m}|$, let $n=m+1$ and let $\kappa>0.$ Then
> $$
> \tau_{t}=\begin{cases}
> \frac{\kappa nt}{|r_{m}|} & \text{if }t\in[0,1/n]\\\\
> \frac{\kappa(nt-1)}{|r_{m-1}|}+\frac{\kappa}{|r_{m}|} & \text{if }t\in[1/n,2/n]\\\\
> \cdots\\\\
> \frac{\kappa(nt-(m-1))}{|r_{1}|}+\kappa\left(\frac{1}{|r_{2}|}+\cdots+\frac{1}{|r_{m}|}\right) & \text{if }t\in[(m-1)/n,m/n]\\\\
> \left(1-\kappa\left(\frac{1}{|r_{1}|}+\cdots+\frac{1}{|r_{m}|}\right)\right)t+\kappa\left(\frac{1}{|r_{1}|}+\cdots+\frac{1}{|r_{m}|}\right) & \text{if }t\in[m/n,1]
> \end{cases}
> $$
>
> Then we have that $p_{i}$ is learnt when $t\in[(m-i)/n,(m-i+1)/n]$ and the $\sigma^{2}$ will be learnt when $t\in[m/n,1],$ giving rise to $m+1$ different phases. In the special case of $|r_{i}|=|r_{i+1}|,$ both $p_{i}$ and $p_{i+1}$ will already be learnt in $[(m-i)/n,(m-i+1)/n]$ so that the phase on the interval $[(m-i+1)/n,(m-i+2)/n]$ is unnecessary. Taking this consideration into account when using the general formula we have derived applied for the two-mode GM gives the time dilation formula from Eq. (10) in our paper. The only difference is that the time dilation here maps $[0,1]$ to $[0,1]$ and in the paper we map $[0,1]$ to $[0,2].$ We included this in the Appendix, Section E, of the revised version of the paper. Note that the idea to detect transitions via comparing the coefficient of data and noise, expressed formally in Lemma 7, is quite general and is a way to extend the time dilation to even more general data distributions.

---

> > ### Comment · Reviewer_19x1 · 2024-11-26
> >
> > Thank you for the detailed revision and response. The updated related work section is very good.
> >
> > I find the new formula for multiple symmetry breaking points to be very interesting. However, I do think that there is some work to be done in integrating it in the paper, which should also include new experiments on this more complex scenario.
> >
> > Therefore, I think that the work would benefit from a thorough revision and a resubmission.

---

### Official Review · Reviewer_DM8L · 2024-11-03

**Soundness:** 1
**Presentation:** 1
**Contribution:** 2
**Rating:** 3
**Confidence:** 3

**Summary:**

This paper investigates the training dynamics of a two-layer autoencoder in a flow-based generative model for high-dimensional Gaussian mixtures. The authors identify and address a phase loss issue in high-dimensional settings, introducing a time dilation technique to ensure sequential learning of high-level and low-level features. This approach enables the model to focus on relevant parameters in each training phase, with experiments on MNIST demonstrating the effectiveness of feature-specific time intervals for improving accuracy.

**Strengths:**

***Relevance of Topic***: The paper addresses a highly relevant and popular topic within the field, focusing on a new approach to velocity field learning in generative models.

***Potentially Valuable Theoretical Contributions***: The theoretical considerations presented in the paper have promise for advancing the understanding of velocity field modeling. If better supported by clear explanations and experimental validation, these theoretical insights could offer a substantial contribution to the committee’s knowledge and help guide future research.

**Weaknesses:**

***Lack of Supported Contributions***: The paper presents several contributions that are not adequately supported by theoretical or empirical evidence. For example, the claim that Θd(1) samples are sufficient to learn the velocity field lacks direct validation. Neither the theoretical analysis nor the experimental results substantiate this assertion, leaving the reader unable to evaluate its validity.

***Clarity and Readability Issues***: The paper is difficult to follow due to an overuse of unexplained symbols and insufficiently detailed explanations of key formulas. These issues hinder the readability of the manuscript, making it challenging to understand the proposed methodology and to fairly evaluate the approach.

***Incomplete Structure and Missing Conclusion***: A major drawback is the absence of a conclusion section, which leaves the paper feeling incomplete. The goals, outcomes, and implications of the methodology and experimental results are not adequately summarized, giving the impression that the paper was prepared hastily.

***Unsupported Claims about the Velocity Field Learning Process***: The claim that the velocity field learns to simplify across phases—for instance, focusing on high-level features in the first phase and low-level features in the second—is not supported by clear theoretical or experimental evidence. This key concept would benefit from illustrative toy examples or empirical analysis, highlighting how these high- and low-level features impact the generation process. Additionally, the meaning of "high-level" and "low-level" features and how these are learned in each phase require further clarification, as the specific factors distinguishing feature extraction in each phase are unclear.

***Narrow Focus on a Specific Architecture***: The proposed network architecture, described in Equation (7), has theoretical justification, but the experiments do not convincingly demonstrate its practical quality or usability. It is also unclear whether the two-phase approach can be generalized to other commonly used architectures, such as U-Net-based models or transformers, which limits the broader applicability of the proposed method.

***Unclear and Incomplete Experimental Evaluation***: The experimental evaluation lacks a well-defined objective, making it difficult to interpret the results in the context of the paper's claims. To strengthen this section, the authors should include:
- Toy examples that visually demonstrate the high- and low-level features associated with each phase of training.
- An analysis of the two-phase training approach using benchmark datasets to assess its impact on data generation quality. This assessment should include both quantitative metrics, such as Fréchet Inception Distance (FID) and Negative Log-Likelihood (NLL), as well as qualitative evaluations to provide a comprehensive view of the method’s performance.

**Questions:**

Please refer to the weakness section.

---

### Official Review · Reviewer_CF3e · 2024-11-03

**Soundness:** 2
**Presentation:** 2
**Contribution:** 1
**Rating:** 3
**Confidence:** 3

**Summary:**

This paper proposes a novel approach to training and analysing diffusion models by splitting the process into two distinct phases. The first phase focuses on determining the mass assignments of clusters to the data, while the second phase is dedicated to learning the structure of each individual cluster. The authors provide explicit calculations for a bimodal Gaussian, which indicate the change point between these two phases. Additionally, experiments are conducted on the MNIST dataset to demonstrate the approach in practice.

**Strengths:**

This paper provides explicit formulas for identifying the phase transition between the two phases described by the authors in a diffusion process. These formulas, although cumbersome to derive, are valuable in the context of the standard bimodal Gaussian example used for sampling. By offering a precise mathematical framework to describe this phase transition, the authors make a contribution for the bimodal Gaussian.

**Weaknesses:**

The main weakness lies in the lack of generalisability of the analysis, and further overall contribution of this work. While the authors provide a detailed investigation of a bimodal example, it remains unclear how this extends to more complex, multimodal data. For datasets with more than two modes, it is not obvious that there are always two phases along the diffusion path, as suggested by the authors. In such cases, there may be multiple occurrences of cluster splitting, and without prior knowledge of the underlying probability density, it is difficult to derive explicit formulas. This limits the broader applicability of the approach to more general probability distributions.

Furthermore, the use of the MNIST dataset, although a recognised benchmark, does not sufficiently demonstrate the generalisability of the authors' claims. To support their conclusions, a more diverse set of examples, ideally involving a variety of multimodal distributions, would provide stronger evidence.

From my current understanding, the presented framework may struggle to accommodate general Gaussian mixtures. If such formulas could indeed be extended, one would expect to see multiple phases emerging during training that cannot be easily identified computationally. This raises questions about the practical utility of the method in more complex settings. Additionally, it is unclear why the phase transitions along the diffusion path are not evident from the mollification process described by the Ornstein-Uhlenbeck SDE, especially for multimodal data well studied in the annealed Langevin literature.

I would encourage the authors to clarify whether the paper aims to claim that, in general, there are two phases to be considered during training for multimodal data, or if this only holds true for bimodal Gaussian mixtures. A clearer articulation of this distinction would significantly aid in understanding the broader relevance of the work. Further elaboration on how these results might generalise to more complex probability densities would also help assess the robustness of the proposed approach.

**Questions:**

Could the authors comment on the expected behaviour for a Gaussian mixture with three clusters? Specifically, if the cluster means are centred at vectors $(-1,...,-1)$, $(0,...,0)$, and $(100,...,100)$, with unit variances and equal weighting across the clusters, would we still observe two distinct phases as described for the bimodal case?

Additionally, could the authors provide further experimental results for multimodal distributions with more than three clusters? A detailed description of the diffusion schedule used in these experiments would also be valuable for better understanding the behaviour of the model in more complex settings.

---

> ### Author Response · Authors · 2024-11-24
>
> Thank you for your detailed review. We address your comments as follows
>
> **P1. For datasets with more than two modes, it is not obvious that there are always two phases along the diffusion path.** Our theoretical argument claiming that a neural network can learn the different phases is _only_ for the two-mode Gaussian Mixture (GM). However, we prove below the existence of more than two phases when the data comes from a GM with more than two modes. (Showing that a neural network can learn more than two phases remains an open problem.) Consider $\mu=\sum_{i=1}^{m}p_{i}\mathcal{N}\left(r_{i},\sigma^{2}\text{I}\right)$ where $r_{i}\in\mathbb{R}^{d}$ and $|r_{i}|$ goes to infinity with $d,$ but $m,p_{i},\sigma^{2}$ are constant with respect to $d.$ If $X_{t}$ is the generative model associated with the interpolant $I_{t}=(1-t)z+ta$ where $z\sim\mathcal{N}(0,\text{I})$ and $a\sim\mu,$ then $X_{t}$ estimates $p_{i}$ at times of the order $1/|r_{i}|.$ We show this in Proposition 2 in the Appendix, Section E, of the revised version of the paper by arguing that it is only at times of order $1/|r_{i}|$ that the denoiser associated to $r_{i}\cdot X_{t}/|r_{i}|$ is nontrivial. Hence, to estimate $p_{i}$ we require a time dilation $\tau_{t}$ such that there exists $a$ and $b$ with $b-a=\Theta_{d}(1)$ where
>
> $$
> \begin{align}
> \tau_{t}=\Theta_{d}\left(\frac{1}{|r_{i}|}\right)\text{ for }t\in[a,b]
> \end{align}
> $$
>
> We specify next a time dilation that for every $i$ would ensure that this condition on $\tau_t$ is fulfilled. Assume $|r_{1}|\leq|r_{2}|\leq\cdots\leq|r_{m}|$, let $n=m+1$ and let $\kappa>0.$ Then
> $$
> \tau_{t}=\begin{cases}
> \frac{\kappa nt}{|r_{m}|} & \text{if }t\in[0,1/n] \\\\
> \frac{\kappa(nt-1)}{|r_{m-1}|}+\frac{\kappa}{|r_{m}|} & \text{if }t\in[1/n,2/n]\\\\
> \cdots\\\\
> \frac{\kappa(nt-(m-1))}{|r_{1}|}+\kappa\left(\frac{1}{|r_{2}|}+\cdots+\frac{1}{|r_{m}|}\right) & \text{if }t\in[(m-1)/n,m/n]\\\\
> \left(1-\kappa\left(\frac{1}{|r_{1}|}+\cdots+\frac{1}{|r_{m}|}\right)\right)t+\kappa\left(\frac{1}{|r_{1}|}+\cdots+\frac{1}{|r_{m}|}\right) & \text{if }t\in[m/n,1]
> \end{cases}
> $$
>
> Then we have that $p_{i}$ is learnt when $t\in[(m-i)/n,(m-i+1)/n]$ and the $\sigma^{2}$ will be learnt when $t\in[m/n,1],$ giving rise to $m+1$ different phases. In the special case of $|r_{i}|=|r_{i+1}|,$ both $p_{i}$ and $p_{i+1}$ will already be learnt in $[(m-i)/n,(m-i+1)/n]$ so that the phase on the interval $[(m-i+1)/n,(m-i+2)/n]$ is unnecessary. Taking this consideration into account when using the general formula we just provided applied for the two-mode GM gives the time dilation formula from Eq. (10) in our paper. The only difference is that the time dilation here maps $[0,1]$ to $[0,1]$ and in the paper we map $[0,1]$ to $[0,2].$ We included this in the Appendix, Section E, of the revised version of the paper.
>
> _Although we believe this sharp asymptotic characterization of times where different parameters are learnt is novel, we stress that the main contribution of our paper is to provide the first theoretical argument showing that a neural network can learn the two phases in the two-mode GM,_ providing the next step to Cui et al 2024 who assume a symmetric two-mode GM. In fact, as far as we are aware it is the first theoretical result linking the choice of training schedule with speciation times. Since in practice the training schedule is just determined by drawing times uniformly, this result could open the door to new and more efficient training methods. It is characteristic of machine learning theory (and specially of new analysis methods) that they will unavoidably deal with simpler settings than those from practice. Indeed, the denoiser for the GM with more than two modes does not have a simple expression like the one for the two-mode GM (see Albergo et al 2023 for the general case.) This makes the analysis much harder, but it is a clear direction for future work.

---

> ### Author Response · Authors · 2024-11-24
>
> **P2. Could the authors comment on the expected behaviour for a Gaussian mixture with three clusters?** You asked about the case where $r_{1}=(-1,\cdots,-1),r_{2}=(0,\cdots,0),r_{3}=(100,\cdots,100)$ where $r_{i}\in\mathbb{R}^{d}$ and $p_{1}=p_{2}=p_{3}=1/3,$ $\sigma^{2}=1.$ The analysis from above yields that $\tau_{t}$ needs to be of order $1/\sqrt{d}$ for an interval of constant length to recover $p_{1}$ and of order $1/(100\sqrt{d})$ in another interval of constant length to recover $p_{3}.$ We note that $|r_{2}|=0$ so that it does not fulfill our assumption that $|r_{2}|$ goes to infinity with $d.$ This assumption is not needed, and the analysis can be extended for general $|r_{i}|.$ However, to understand your question this is not needed: indeed, estimating $p_{1}$ and $p_{3}$ correctly ensures correct estimation of $p_{2}$ since these three numbers add to $1$. We hence have three phases in your proposed problem: for times of order $1/(100\sqrt{d})$ where the parameter $p_{3}$ is estimated, for times of order $1/\sqrt{d}$ the parameter $p_{1}$ (and hence $p_{2}$) is estimated, and for times of order $1$ the paramter $\sigma^{2}$ is estimated.
>
> **P3. There may be multiple occurrences of cluster splitting, and without prior knowledge of the underlying probability density, it is difficult to derive explicit formulas. This limits the broader applicability of the approach to more general probability distributions**. It is true that for a general probability distribution, we cannot derive an explicit formula for the time where the phase transition occurs. However, explicit formulas are not needed to determine these times. Indeed, in Section 6.2 of the original paper we provided a practical approach for determining this time for any probability distribution in terms of the U-Turn and exemplified the method for the MNIST dataset. Does this address your concern?
>
> **P4. Additionally, it is unclear why the phase transitions along the diffusion path are not evident from the mollification process described by the Ornstein-Uhlenbeck SDE.** It is worth clarifying that we are not considering the Ornstein-Uhlenbeck SDE but instead we are working under the framework of the probability flow ODE, so that given the initial condition there is no noise added in the generation process. On the other hand, it is True that the phase transition times can be determined by looking at the ratio of the data and noise coefficients of a noisy version of the data $I_{t}=(1-t)z+ta$ where $z$ is noise and $a$ is data (this is shown formally in Lemma A3 below). We used this idea implicitly for our results, but we will add this explicit formulation from Lemma A3 to the paper. Is this what you meant by the phase transitions being evident from mollification process?

---

### Official Review · Reviewer_j664 · 2024-11-04

**Soundness:** 1
**Presentation:** 1
**Contribution:** 1
**Rating:** 1
**Confidence:** 2

**Summary:**

This paper presents a new training approach for flow-based generative models designed to overcome the challenge of learning high-level features in high-dimensional spaces. The authors introduce phase-aware strategy combined with a time dilation mechanism to provide an appropriate time schedule for the model to capture high-level structures. Using a two layer autoencoder and a two-mode Gaussian Mixture Model, they show that the neural network representing the velocity field learns to simplify only phase-relevant parameters. Validation using the MNIST dataset indicates that their method identifies a time interval during which additional training significantly enhances accuracy for specific features.

**Strengths:**

1. The paper has explained in depth the mathematical background of the method.
2. It showed an interesting idea.

**Weaknesses:**

1. The standalone paper does not show any experiments or proof of them.
2. The paper does not include a comparison with other state-of-the-art methods.
3. The paper should include clear picture with network architecture.
4. It is almost only theoretical without clearly presenting the necessary experiments or methods that would be expected.
5. The paper is hard to follow and has some weirdly written sentences, i.e. “”Sample complexity for Gaussian Mixtures Cui et al. (2024) study the learning problem for the Gaussian mixture in high dimensions demonstrate n = Θd(1) samples are sufficient to sample the in the balanced case where (the two modes have the same probability.)”
6. The paper lacks quantitative and qualitative comparisons. Two graphs included show more of an ablation study and not the comparison.
7. It has a well-written theoretical side but without any visual or quantitative results, it is not sufficient to defend the theory.

**Questions:**

1. Do you have a visual comparison of the MNIST dataset? If so, can you include it in the main paper?
2. Did you do a comparison with state-of-the-art methods? Can you please include it in the tables and figures in the main paper?
3. Could you please check the grammatical correctness of the paper? Some sentences have unnecessary brackets and overall the paper is hard to follow and understand.
4. Could you please include a figure showcasing your proposed method? It would enhance the readability of your paper.

---

> ### Author Response · Authors · 2024-11-25
>
> Thanks for your review. We address your comments as follows.
>
> **W1. The standalone paper does not show any experiments or proof of them.** The original paper has experiments in Section 6.
>
> **W3. The paper should include clear picture with network architecture.** The architecture we analyze is shown in Equation 8. Since it is a very simple two-layer Denoising Autoencoder, we believe that showing the formula mathematically is best to describe it. On the other hand, the architecture used for the experiments is described in the Appendix, Section D, and is also simple. We stress that these are standard network architectures widely used by previous work.
>
> **W4. It is almost only theoretical without clearly presenting the necessary experiments or methods that would be expected.** ICLR is a multidisciplinary conference that regularly accepts papers that are purely theoretical. What experiments or methods are you referring that would be expected?
>
> **W5. The paper is hard to follow and has some weirdly written sentences.** We fixed the typo in the sentence you mentioned and went through the paper to check its grammatical correctness. Is there anything else in the paper that makes it hard to follow?
>
> **W6. The paper lacks quantitative and qualitative comparisons. Two graphs included show more of an ablation study and not the comparison.** We note that both of our experiments show the benefit of dilating time by comparing the dilated interpolant with the non-dilated interpolant. What comparison do you have in mind?
>
> **W7. It has a well-written theoretical side but without any visual or quantitative results, it is not sufficient to defend the theory.** We provide quantitative results in Section 6.
>
> **Q1. Do you have a visual comparison of the MNIST dataset? If so, can you include it in the main paper?** In the Appendix, Section C2, we show samples from MNIST using the dilated interpolant and the non-dilated interpolant. Does this address your concern about a visual comparison of the MNIST dataset?
>
> **Q3. Could you please check the grammatical correctness of the paper? Some sentences have unnecessary brackets and overall the paper is hard to follow and understand.** We went through the paper and checked for its grammatical correctness. Please let us know if the revised version has anything that makes it hard to follow.

---

> > ### Comment · Reviewer_j664 · 2024-11-26
> > **Comment by Reviewer**
> >
> > Thank you for your answer, but I still think the paper needs improvement. I stay with my score.

---

### Meta-Review · Area_Chair_n5o5 · 2024-12-19

**Metareview:**

This paper studies the learning dynamics of generative models through carefully analyzing a controlled and simplified setting - a diffusion model consisting of "a two-layer autoencoder used to parameterize a flow based generative model for sampling from a high-dimensional Gaussian mixture".  The authors identify two phases of learning, which help to better understand the training dynamics and simplify learning.  Note, one review appears to be potentially LLM generated, so I am discounting that review.  The remaining reviewers found the problem well motivated and relevant to the community.  However, the reviewers all voted to reject the paper, even after reading the author response.   They questioned how well the analysis would generalize to more common models, wanted more experiments demonstrating the findings, and in general found the paper difficult to follow.  Note, one reviewer seemed much more positive about the paper following the authors' response.  However, they felt the paper needed too much work still to vote for accept.  Perhaps integrating that response into the manuscript, along with careful editing, would make for a much stronger submission to a future venue.

**Additional Comments On Reviewer Discussion:**

Initial reviews were quite negative, owing to issues with clarity and concerns about whether the results would generalize to more complex settings.  One of the reviewers also found the related work section lacking.  After reading the response, one reviewer elected to keep their score.  Another found the new related work section very well written, and the new theoretical arguments compelling.  However, they felt the paper needed too much work still to vote for accept.

---

### Decision · Program_Chairs · 2025-01-22

Reject